# Changes in Andes Mountains snow cover from MODIS data 2000-2016

Freddy A. Saavedra[1,2], Stephanie K. Kampf[3], Steven R. Fassnacht[3], and Jason S. Sibold[4]

[1] Departamento de Ciencias Geográficas, Facultad de Ciencias Naturales y Exactas, Universidad de Playa Ancha, Leopoldo Carvallo 270, Playa Ancha, Valparaíso, Chile.

[2] Centro de Estudios Avanzados, Universidad de Playa Ancha. Traslaviña 450, Viña del Mar, Chile.

[3] Department of Ecosystem Science and Sustainability, Colorado State University.

[4] Department of Anthropology, Colorado State University.

*Correspondence to*: Freddy Saavedra (freddy.saavedra@upla.cl)

**Abstract.** The Andes Mountains span a length of 7,000 km and are important for sustaining regional water supplies. Snow variability across this region has not been studied in detail due to sparse and unevenly distributed instrumental climate data. We calculated snow persistence (SP) as the fraction of time with snow cover for each year between 2000-2016 from Moderate Resolution Imaging Spectroradiometer (MODIS) satellite sensors (500 m, 8-day maximum snow cover extent). This analysis is conducted between 8° S and 36° S due to high frequency of cloud (>30% of the time) south and north of this range. We ran Mann-Kendall and Theil-Sens analyses to identify areas with significant changes in SP and snow line (the line at lower elevation where SP=20%). We evaluated how these trends relate to temperature and precipitation from Modern-Era Retrospective Analysis for Research and Applications-2 (MERRA2) and University of Delaware datasets) and climate indices as El Niño Southern Oscillation (ENSO), Southern Annular Mode (SAM), and Pacific Decadal Oscillation (PDO). Areas north of 29° S have limited snow cover, and few trends in snow persistence were detected. A large area (34,370 km$^2$) with persistent snow cover between 29-36° S experienced a significant loss of snow cover (2-5 fewer days of snow year$^{-1}$). Snow loss was more pronounced (62% of the area with significant trends) on the east side of the Andes. We also found a significant increase in the elevation of the snow line at 10-30 m year$^{-1}$ south of 29-30° S. Decreasing SP correlates with decreasing precipitation and increasing temperature, and the magnitudes of these correlations vary with latitude and elevation. ENSO climate indices better predicted SP conditions north of 31° S, whereas the SAM better predicted SP south of 31° S.

## 1 Introduction

The influence of a changing climate on snow cover has been studied across the world (Barnett et al., 2005; Dahe et al., 2006; Masiokas et al., 2006; Adam et al., 2009; Brown and Mote, 2009). In snow-dominated basins, snowpack often provides the largest reservoir of water (Masiokas et al., 2006; Adam et al., 2009), which influences stream discharge, affecting erosion, sediment transport, hydropower production, and potential water storage (Hall et al., 2012; Stewart, 2009; Arsenault et al., 2014). Peru, Bolivia, Chile, and Argentina all depend on snow and/or glacier melt for water supply due the short rainy seasons (Bradley et al., 2006; Peduzzi et al., 2010; Masiokas et al., 2010; Rabatel et al., 2013). These basins are sensitive to climate changes that modify either temperature or precipitation. In general, increasing temperatures cause decreases in snow cover by increasing the elevation of the 0 °C isotherm in mountain regions and/or decreasing the fraction of precipitation falling as snow (Barnett et al., 2005; Brown and Mote, 2009). The magnitude of this effect decreases with increasing elevation (López-Moreno et al., 2009). However, snow cover changes can be difficult to predict in areas where temperature

increases are accompanied by changes in precipitation (Dahe et al., 2006; Adam et al., 2009). Moreover, decreases in the duration of snowpack cause negative feedbacks due to changes in albedo, which leads to increased absorption of solar radiation, intensification of warming trends, and further reductions in snowpack (Fassnacht et al., 2016).

In the Southern Hemisphere, seasonal snow cover is primarily confined to southern South America. In tropical latitudes (20° S), the inter-tropical convergence zone (ITCZ) extends southward during the austral summer (January and February) to produce much of the annual precipitation in Altiplano regions (20° S) (Barry, 2008). Further south (17° S to 31° S), hyper-arid condition lead to no presence of seasonal snow even at high elevations (>5,000) (Saavedra et al., 2017) and glaciers above 6,700 m (Hobbs et al., 1999). In extratropical areas (31–35° S), higher precipitation and lower temperatures lead to extensive austral winter snow cover (Garreaud, 2009; Foster et al., 2009), and strong connections have been identified
between precipitation and seasonal snow cover in the central Chilean Andes (33° S) (Pellicciotti et al., 2007). South America's precipitation patterns are influenced by the El Niño Southern Oscillation (ENSO), with above average annual precipitation during El Niño events and below average totals La Niña events in central latitudes (30° S to 37° S) (Rutllant and Fuenzalida, 1991; Mernild et al., 2017; Garreaud et al., 2009; Gascoin et al., 2011). The interannual seasonal variability of precipitation in South America has been related to the Southern Annular Mode (SAM) (Vera and Silvestri, 2009; Fogt et
al., 2010) and the Pacific Decadal Oscillation (PDO).

Air temperature increases ($\sim$0.25 °C decade$^{-1}$), have been documented for the central Andes between 1975 and 2001 (Falvey and Garreaud, 2009), and this has led to a rise of the 0 °C isotherm by $\sim$120 m in winter and $\sim$200 m in summer in the same area (Carrasco et al., 2008). However, the impacts of climate change on snow covered areas throughout South America have not been studied in detail due to sparse and unevenly distributed climate data (Masiokas et al., 2006; Aravena and Luckman,
2009; Cortés et al., 2011). Analyses of 17 surface stations document a non-significant positive linear trend in snow water equivalent (SWE) between 30° and 37° S from 1951-2007 (Masiokas et al., 2010), with snow patterns strongly linked to year-to-year climatic oscillations (Masiokas et al., 2012). Relatively uncertain or biased input forcing data for the region have limited the ability to conduct snow analyses (Cortés and Margulis, 2017), leaving open questions about patterns of snow accumulation and controls on its inter-annual variability throughout the Andes (Cortes et al. 2016).

Recent studies have begun to combine remote sensing information with the ground monitoring network to expand the geographic scope of snow studies in the Andes. Fractional snow covered area information from satellite sensors has been combined with energy balance modelling to reconstruct snow water equivalent in an area from 28-37° S (Cornwell et al., 2016) and in select basins between 30-34° S (Cortés et al., 2014; Ayala et al., 2014). Across the Andes as a whole, snow modelling has been conducted to estimate snow patterns from 1979-2014 (Mernild et al., 2017), but the sparse in-situ
networks that limit snow trend studies also cause problems for the data used as model inputs over the Andes Mountains (Cortés et al., 2016; Yi et al., 2011). Therefore, while modelling studies provide useful information about likely snow patterns, there is still a need for snow observations throughout the region to evaluate how well the models simulate snow patterns. This study is the first observation-based study of snow spatial and temporal patterns across the Andes region, contributing to the open questions about snow accumulation patterns highlighted by Cortes et al. (2016). We use snow cover
information from the Moderate Resolution Imaging Spectroradiometer (MODIS) to (1) quantify changes in snow cover and their spatial patterns associated with latitude and elevation in the Andes, and (2) assess how these snow cover changes relate to climate variables.

## 2 Materials and Methods

### 2.1 Study Site

The Andes Mountains cross seven countries (Venezuela, Colombia, Ecuador, Peru, Bolivia, Chile, and Argentina) and a wide range of latitudes (10º N to 57º S) (Figure 1). They represent the highest mountain system outside of Asia and are the longest (7,000 km) mountain chain in the world (Barry, 2008). The Andes have a strong effect on atmospheric circulation (Llamedo et al., 2016), and local climates in the Andes region vary greatly depending on latitude, elevation, and proximity to the ocean (Garreaud et al., 2009).

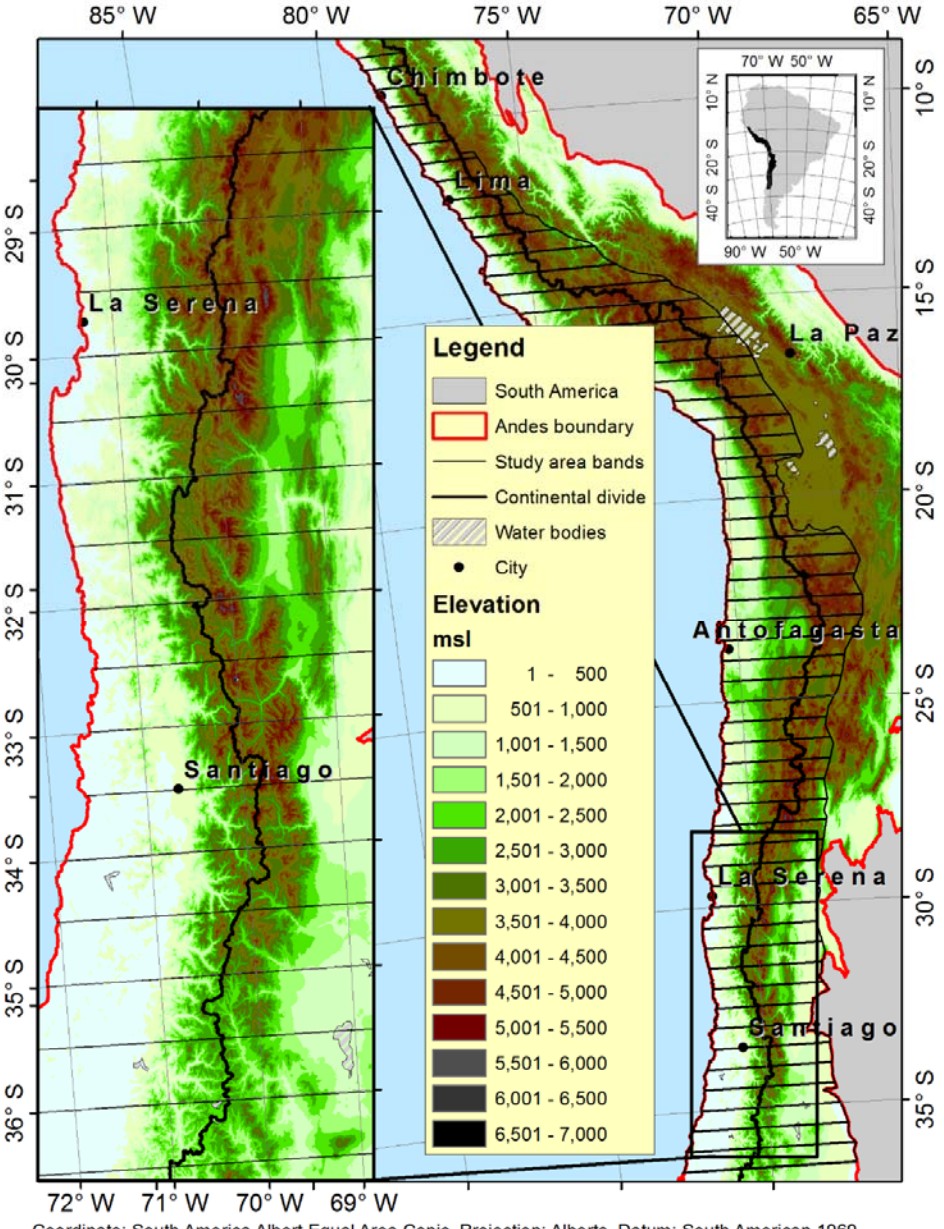

Figure 1. Study area (red line) over digital elevation model of the Central Andes Mountains subdivided into 0.5° latitude bands (black lines). The zoomed in image highlights the area with more snow.

According to the Modern-Era Retrospective Analysis for Research and Applications-2 (MERRA2) dataset 2000-2016, precipitation is higher on the windward side (east in the north, west in the south) of the Andes due to the orographic effect (Garreaud et al., 2009; Quintana, 2012) (Figure 2a). In the northern area (north of 20° S), precipitation is concentrated in the austral summer (DJFM), with higher precipitation on the east side. Areas between 20-30° S have hyper-arid conditions with limited precipitation all year on the west side and greater precipitation on the east side. In this area, wet episodes tend to occur during the austral summer due to strong upper-level easterly winds that enhance moisture transport from Amazonia (Garreaud et al., 2003). South of 30° S, precipitation has a well-defined annual pattern with a peak of precipitation in austral winter (JJA) (Quintana, 2012; Valdés-Pineda et al., 2015a). At these latitudes, westerlies from the Pacific bring moist air masses inland, and the west side of the Andes receives more precipitation than the east side (Garreaud et al., 2009; Matsuura and Willmott, 2015).

Throughout the Andes, the lowest mean temperatures are during the austral winter (JJA), and seasonal temperature differences increase with latitude (Figure 2b). Mean annual temperature decreases from north to south and with increasing elevation. The combined precipitation and temperature patterns affect snow cover duration in the region. Snowfall patterns have been mapped in detail for this region using remote sensing, but high cloud cover (>30% of the time) limited this mapping to the area between 8° S and 36° S (Saavedra et al., 2017). Saavedra et al. (2017) used the Moderate Resolution Imaging Spectroradiometer (MODIS) snow cover product (MOD10A2) to map the monthly average snow covered area (SCA) from 2000 to 2014 for latitude bands across the Andes (Figure 2c). The largest snow covered areas and longest duration snow cover (>30% of year) is south of 24° S. On the west side between 24-33° S, the snow season lasts from around day 110 (April 20th) to day 280 (October 7th). Between 33-35° S snow cover can persist all year in high elevation areas (>5000 m). South of 36° S, the snow season also starts around day 110 but lasts longer into the austral spring until around day 320 (November 16th). The latitudinal variability of snow cover can be summarized using snow persistence (SP), which is the fraction of a year with snow cover. North of 25° S, average snow persistence is lower than 10%, and the snow line snow line (defined SP=20%) is over 5000 m (Figure 2d) (Saavedra et al., 2017). The snow line decreases in elevation with increases in latitude south of 25° S with a consistently lower snow line on the west side than on the east side of the range (Vuille and Ammann, 1997; Williams and Ferringo, 1998; Barry and Seimon, 2000; Saavedra et al., 2017).

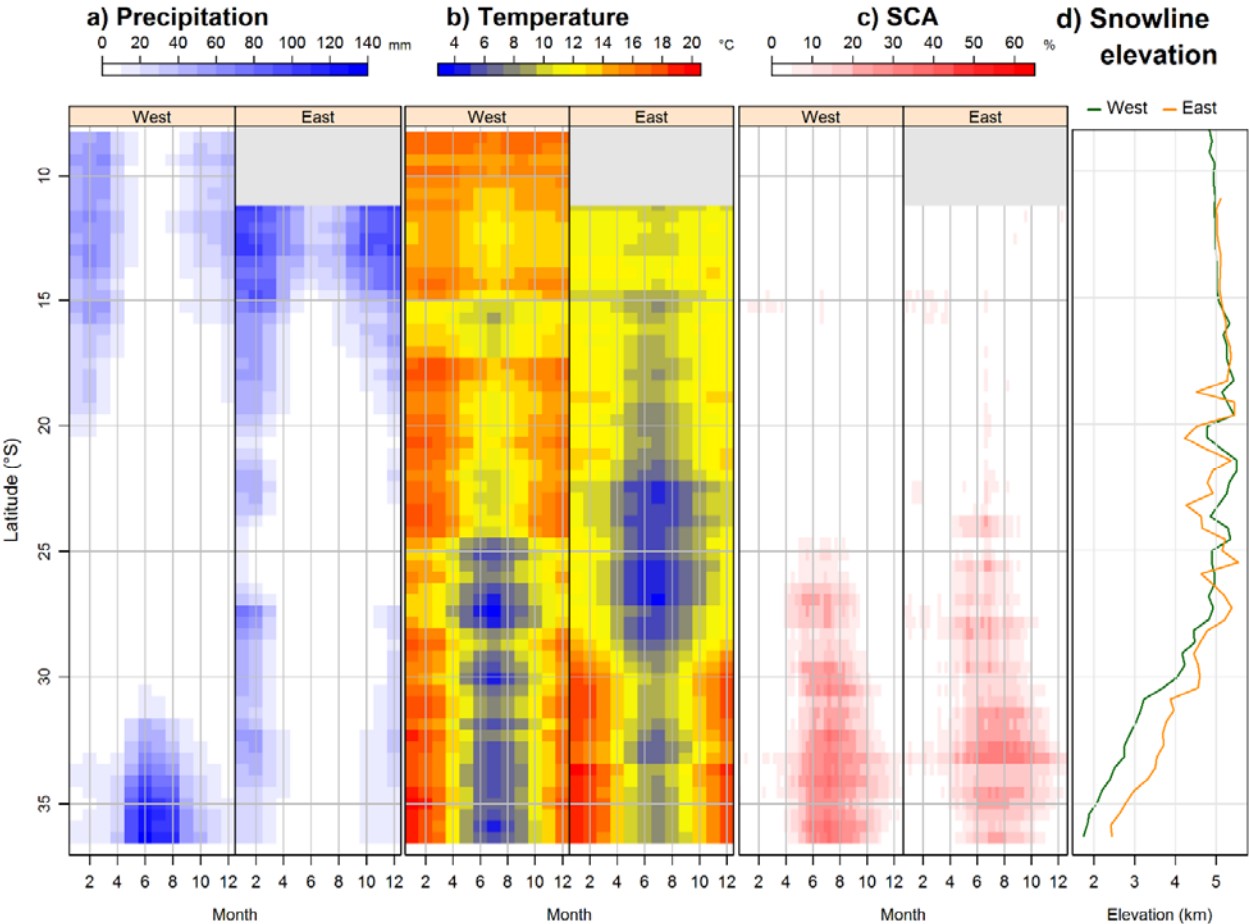

Figure 2. Mean monthly hydroclimate variables by side of the continental divide (west/east) and latitude band for the period 2000-2016 for a) precipitation and b) temperature from the MERRA2 dataset, and c) fraction of area with snow cover calculated from the binary 8-day product from MODIS (MOD10A2). The d) snow line elevation (SP=20%) was derived using the methods of (Saavedra et al., 2017). Grey areas represent latitudes masked due to high frequency of cloud cover in snow covered area analysis (>30% time period).

## 2.2 Data

We used the MODIS eight-day 500m binary snow cover products Collection 5 Level 3 (Riggs et al., 2006). MODIS is a passive 36-band spectrometer on board two satellites, Terra and Aqua (Hall et al., 2002). One of the spectral bands used to calculate the snow products for the Aqua satellite has malfunctioned, so our research is based on Terra products. Due to the frequent presence of clouds in the daily MODIS snow cover products, we used the eight-day maximum product (MOD10A2), which represents the maximum snow cover and minimum cloud during each eight-day interval (Riggs et al., 2006). The study area is covered by seven MODIS images (tiles), which we downloaded for the time period from 2000 to 2016 (MOD10A2) (http://reverb.echo.nasa.gov), giving a total of 5,833 tiles of MOD10A2 (Hall et al., 2006).

We retrieved monthly precipitation (P) and mean temperature (T) from the MERRA2 dataset for 1980 - 2016 at 0.625°×0.5° longitude/latitude grid resolution (https://gmao.gsfc.nasa.gov/reanalysis/MERRA-2/data_access/). The MERRA2 model has shown good performance in numerical weather prediction, climate reanalysis, and global mesoscale climate models (Molod et al., 2015). We extended the analysis of P and T back to 1960 by combining MERRA2 data for 1980-2016 with the University of Delaware dataset version 4.01 (UDelv4), which has a grid resolution of 0.5° for the 1960-1979 period (Legates

and Willmott, 1990) (http://www.esrl.noaa.gov/psd/data/gridded/data.UDel_AirT_Precip.html). For the study area, UDelv4 compiled data from instrumental ground stations (35 stations across the study area) at a monthly time step from several sources including Global Historical Climatology Network Monthly, Daily Global Historical Climatology Network, Global Synoptic Climatology Network, and Global Surface Summary of Day. The monthly averages of station data were

interpolated to a 0.5° x 0.5° latitude/longitude grid using the spherical version of Shepard's distance-weighting method. In addition, station-by-station cross-validation was employed to evaluate the spatial interpolation errors (Willmott and Matsuura, 2005).

An important source of interannual climate variability in the region is El Niño Southern Oscillation (ENSO) (Masiokas et al., 2012; Cortés et al., 2016). We retrieved 3-month running mean values of four indices (MEI, SOI, ONI, and TNI) and four

sea surface temperature (SST) regions (1+2, 3, 3.4, and 4) that describe ENSO behavior. These values are available at a monthly time step from https://climatedataguide.ucar.edu/climate-data. The multivariate ENSO Index (MEI) is based on the six main observed variables (sea-level pressure, zonal surface wind, meridional surface wind, sea surface temperature, surface air temperature, and total cloudiness fraction of the sky) over the tropical Pacific (http://www.esrl.noaa.gov/psd/enso/mei/table.html) and has been described as a more complete and flexible description of

the ENSO phenomenon (Wolter and Timlin, 1998). The Southern Oscillation Index (SOI) is a standardized index based on the observed sea level pressure differences between Tahiti and Darwin, Australia. The Oceanic Niño Index (ONI) calculates the anomalies of SST in Niño region 3.4 and is used as the standard index to defined Niño/Niña events by National Oceanic and Atmospheric Administration (NOAA). The Trans-Niño Index (TNI) is given by the difference in normalized anomalies of SST between Niño 1+2 and Niño 4 regions. The Niño regions correspond to ship tracks and are defined as Niño 1+2 (0-

10° S, 90-80° W), Niño 3 (5° N - 5° S, 150-90° W), Niño 3.4 (5° N - 5° S, 170-120° W), and Niño 4 (5° N - 5° S, 160° E - 150° W).

Additionally, we explored the potential influence of the Pacific Decadal Oscillation (PDO) on snow persistence (Dettinger et al., 2001; Masiokas et al., 2010) using the values available at https://www.ncdc.noaa.gov/teleconnections/pdo/. PDO is described as a long-lived El Niño-like pattern of Pacific climate variability (Zhang et al., 2016). We also evaluated the

Southern Annular Mode (SAM) because it affects South American precipitation by changing the position of the westerly wind belt that influences the strength and position of cold fronts and mid-latitude storm systems (Gillett et al., 2006). We used the values available at http://www.esrl.noaa.gov/SAM/. SAM is a largely zonally-symmetric mode of variability in the Southern Hemisphere computed from sea level pressure (Marshall, 2003).

We used a digital elevation model (DEM) developed based on Shuttle Radar Topographic Mission (SRTM) (at a 90-m

resolution) and completed with Advanced Spaceborne Thermal Emission and Reflection Radiometer (ASTER) (at a 30-m resolution) elevation data (http://reverb.echo.nasa.gov/) in areas with missing SRTM values. We divided the study area into 100 m elevation bands and 0.5° (approx. 50 km) latitude bands (Saavedra et al., 2017). We changed the spatial resolution of MERRA2 and UDelv4 dataset resampling to 500m to fit with MOD10A2 spatial resolution. The continental divide was defined by using the drainage side (East or West) of the Andes determined from the Watershed Tool in ArcGIS®. All

geospatial data were mosaicked and projected into the South American Albers equal area azimuthal projection to minimize shape distortion (Kennedy and Kopp, 1994).

**2.3 Analysis**

To document snow cover patterns and their changes over time from 2000 through 2016, we calculated the annual snow persistence (SP) for each 500 x 500 m pixel in the study area as the fraction of the images in a year with snow cover (Saavedra et al., 2017). We masked areas with mean annual SP<7% to avoid potential misclassifications in MODIS snow products (Figure 2a) (Hall et al., 2002). This threshold excludes the little to no snow zone (SP<7%) as defined in Saavedra et al. (2017). For the remaining pixels we conducted all geo-statistical analyses using the statistical computing R software (RCoreTeam, 2013). We used the mean annual SP time series 2000-2016 to capture the spatial-temporal variability of snow presence (supplementary material 1a). We used the non-parametric Mann-Kendall analysis to test for trend significance in annual SP (Khaled and Ramachandra, 1998) and quantified the rate of change using the linear Theil-Sen's slope, which determines the slope as the median of all possible slopes between data pairs (Theil, 1950; Sen, 1968). We ran the Mann-Kendall analysis using the "Kendall" package for R (McLeod, 2011). Trends were considered significant at a p-value ≤ 0.05. We also calculated a standardized rate of change by dividing the original Theil-Sen's slope by the mean annual SP. This allowed us to compare the rate of SP change (slope) in different snow zones. For each year, we used the annual SP to calculate the line with SP=20% (our definition of snowline). Then we applied a buffer function of 100 m over this line to create a band of 200 m width and calculated its mean elevation from the DEM (Saavedra et al., 2017). We evaluated the trends in snow line elevation for each 0.5° latitude band on the west and east sides using Mann-Kendall analysis and calculated rates of change by Theil-Sen's slope. Finally, we calculated trends in SP for individual months and elevation bands and examined how the magnitude of trends varied seasonally and with elevation across the study area.

To explore how climate variables were related to snow persistence trends we ran the same trend analyses described for SP on P and T to capture the spatial-temporal variability (supplementary material 1b and 1c). Additionally, we evaluated the linear correlation between each climate variable (P, T, ENSO, PDO, and SAM at monthly and annual time scales) and SP using Pearson's correlation coefficient ($r$). For the climate indexes (ENSO, PDO, and SAM), we examined how the index correlations with SP varied across the study area. To do this, we computed the absolute value of $r$ for each index value correlation with SP. Using only significant correlations, we then computed the sum of the absolute values of $r$ for all pixels in each of the snow regions defined by Saavedra et al. (2017) (Tropical 1: 9-13° S, Tropical 2: 13-24° S, Middle Latitude 1: 24-31 °S, and Middle Latitude 2: 31-36° S). A greater value of this sum indicates higher correlations with SP across the region; for each region we identified the index that best predicts SP as the index greatest sum of absolute values of $r$.

Finally, we ran multiple linear regressions at an annual time step to define the proportion of variability of SP explained by each climate variable (P, T). The strength of these regressions was quantified using the coefficient of determination ($r^2$). Lastly, we calculated the relative importance of independent variables on annual SP using de Lindeman, Merenda and Gold (lmg) approach included in the relative importance for multiple linear regression "relaimpo" R package (Groemping and Matthias, 2013). The lmg method evaluates the individual contribution of each regressor to the full $r^2$ of the model (Grömping, 2006). We visualized all statistical analyses by pixel mapping the geographic distribution of statistical output and latitude-elevation charts to highlight the relationship with latitude and elevation ranges.

## 3 Results

### 3.1 Annual snow persistence and snow line trends

The mean annual snow persistence across the Andes Mountains varies with both latitude and elevation (Figure 3a). The total area with either intermittent, seasonal, or persistent snow (mean annual SP≥7%) is 176,284 km$^2$ and is distributed in four main zones (Saavedra, 2016; Saavedra et al., 2017). The northern area (Tropical 1, 9-13° S) has just 2% of this snow area (3,525 km$^2$); latitudes 13-24° S (Tropical 2) contain 12% (21,154 km$^2$) of the snow area; latitudes 24-31° S (Middle latitude 1) contain 41% (72,277 km$^2$), and the remaining 45% of snow covered areas are all between 31-36° S (Middle latitude 2, 79,328 km$^2$). Mann-Kendall trend analyses of annual SP (Figure 3b) show significant decreasing trends south of 29° S, where 2-5 fewer days of snow every year (-1.5 to -0.5 % year$^{-1}$) are the most common values. Small areas registered an increase of SP (blue range colors in Figure 3b) south of 34° S at lower elevations. The standardized rate of change shows the magnitude of the rate of change normalized by mean annual SP (Figure 3c). North of 34° S the standardized rate of changes is elevation-dependent on both sides of the mountains, with greater changes (more negative trends) at lower SP values. South of 34° S the relative changes in SP are greater (more negative) on the east side than on the west side of the Andes. As a result of the short period of study, we computed the trend of cloud cover to evaluate whether changes in the frequency of clouds could have generated spurious trends. We did not detect any significant trends in cloud frequency in the study area (data not shown), indicating that trends in cloud impairment would likely not have influenced the detected SP trends.

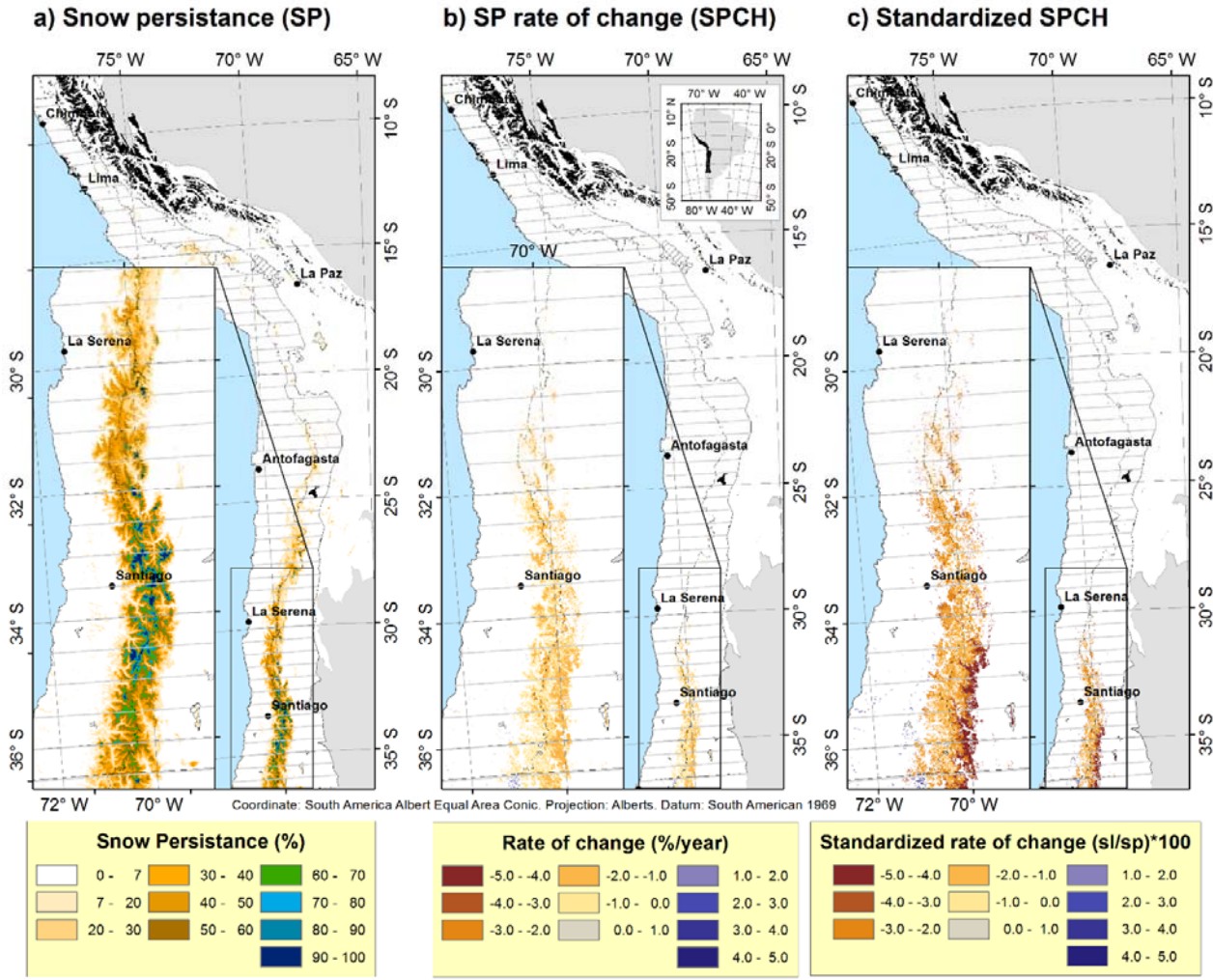

Figure 3. Spatial patterns of a) snow persistence (SP) 2000-2016, b) rate of change (Theil-Sen's slope) in annual SP 2000-2016, and c) standardized rate of change (Theil-Sen's slope/ SP); pixels are only colored in b and c if the trend is significant at p≤0.05. All images exclude areas with little to no snow (mean annual SP≥7%). Areas with frequency of cloud greater that 30% are masked in black.

The latitude-elevation band analysis of SP reveals how the pattern of snow varies with elevation across the region (Figure 4a). North of 23° S the snow is confined to over 5000 m, with a steep change of SP with elevation. South of 26° S, areas with similar SP values are found at lower elevations with increasing latitude. The west side has consistently lower elevation snow than the east side for any value of SP under 90% (Figure 4d). For a more detailed description of these patterns see (Saavedra et al., 2017). North of 19° S, decreasing SP is confined to over 5000 m on both sides of the Andes (Figure 4b). South of 29° S, areas with declining SP occur at lower elevation and vary between west and east sides. The west side has a lower elevation of each category of Theil-Sen's rate of change compared to the east side at the same latitude. The maximum rates of decreasing SP are located at middle-high elevations, between 4000-5000 m at 29° S and at 3000-4000 m at 36° S for both sides (Figure 4b). The maximum rates of SP decline are in the seasonal snow zone, which is defined as mean annual SP between 30-90% (Figure 4e) (Saavedra et al., 2017). In contrast, the standardized Theil-Sen's rate of change shows greater declines at lower values of SP on both sides, except in a few anomalous areas (Figure 4c). This means that the greatest relative declines in SP occur in the intermittent snow zone (Figure 4f).

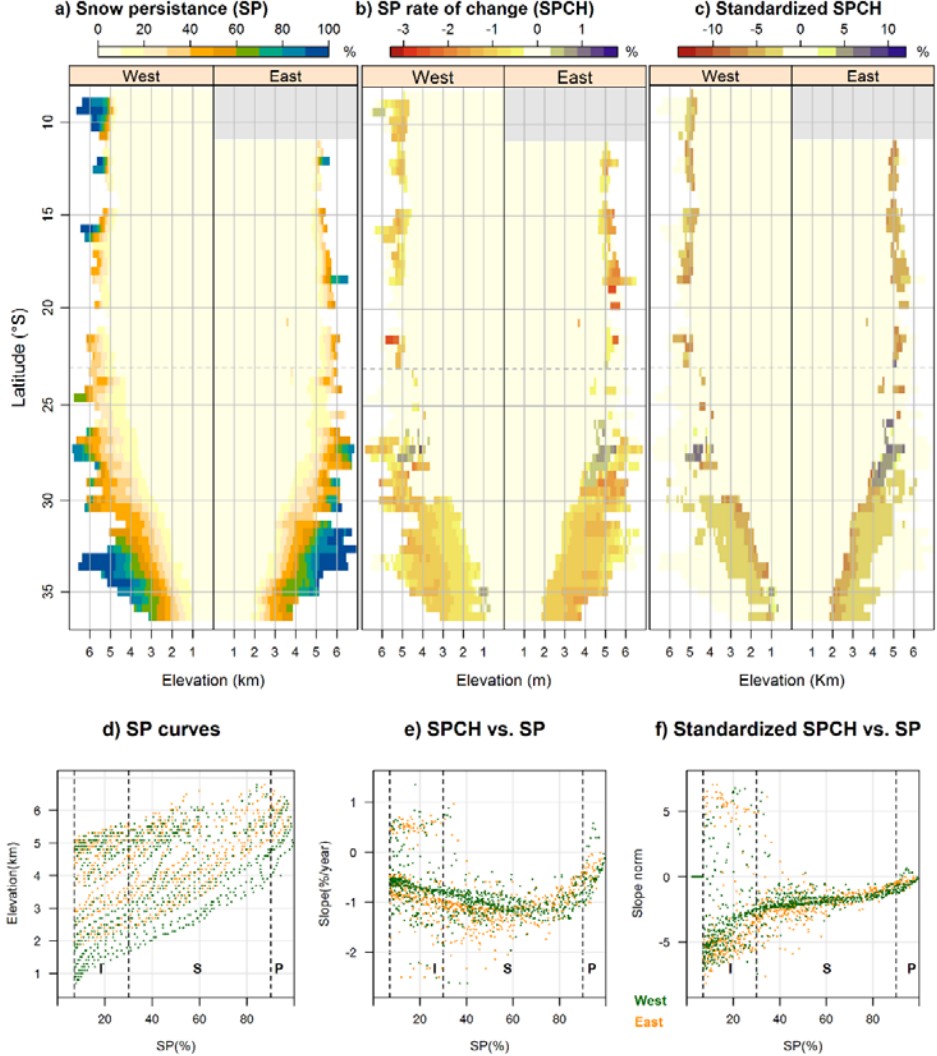

Figure 4. Latitude-elevation band analysis by side of the continental divide (west/east) for a) mean annual SP, b) rate of change (Theil-Sen's slope) from the Mann-Kendall trend test for annual SP, and c) standardized rate of change from 2000 to 2016; only areas with significant trends (p≤0.05) and SP≥7% are colored. Grey areas in a, b, and c, represent latitudes masked due to high frequency of cloud cover in snow covered area analysis (>30% time period). The SP curves in plot (d) show the relation of SP with elevation. The relation between SP and Theil-Sen slope is shown in e), and standardized Theil-Sen slope in f). West (green) and East (orange) sides are plotted in d, e, and f; vertical dashed gray lines represent thresholds for intermittent (I), seasonal (S), and persistent (P) snow zones (Saavedra et al., 2017).

The amount of area affected by significant changes in snow persistence also shows a strong difference between the west and east sides of the Andes (Figure 5a). South of 29° S the total area with Thiel-Sen slopes lower (more negative) than -0.5 (% year$^{-1}$) is 34,370 km$^2$. The majority (62%) of this area is on the east side of the Andes. The most common change values are on the order of -2.0 to -0.5% year$^{-1}$. For areas with the steepest declining trends in SP (-3.0 to -2.5% year$^{-1}$), 79% of the affected area is east of the continental divide (Figure 5d). Trends in snow line elevation (elevation of 20% SP) were only significant south of 29° S on the west side and south of 30° S on the east side of the range (Figure 5b), where the snow line increased in elevation at a rate of about 10-30 m year$^{-1}$. Rates of increase in snow line elevation are higher for the west side than for the east side between 30-32° S, but south of 34° S the east side snow line elevation changes are higher and significant compared with a non-significant trend on the west side.

North of 23°_S the greatest changes in snow persistence were in austral fall (March to May, months 1-3 in Figure 5c) on both side of the Andes (Figure 5e - Tropical). South of 23°_S, the largest changes were in late austral winter (August, month 8 in Figure 5c) for the intermittent snow zone (Figure 5e – Extra-tropical), where no major differences in the seasonality of change between west and east sides were observed. For the seasonal snow zone, the largest changes in snow persistence were during the melting season in austral late winter – early spring (August to October, months 8 to 10 in Figure 5c). Finally, at the upper boundary of the seasonal snow zone and into the persistent snow zone (SP>70%), the largest changes in snow persistence were observed during austral fall.

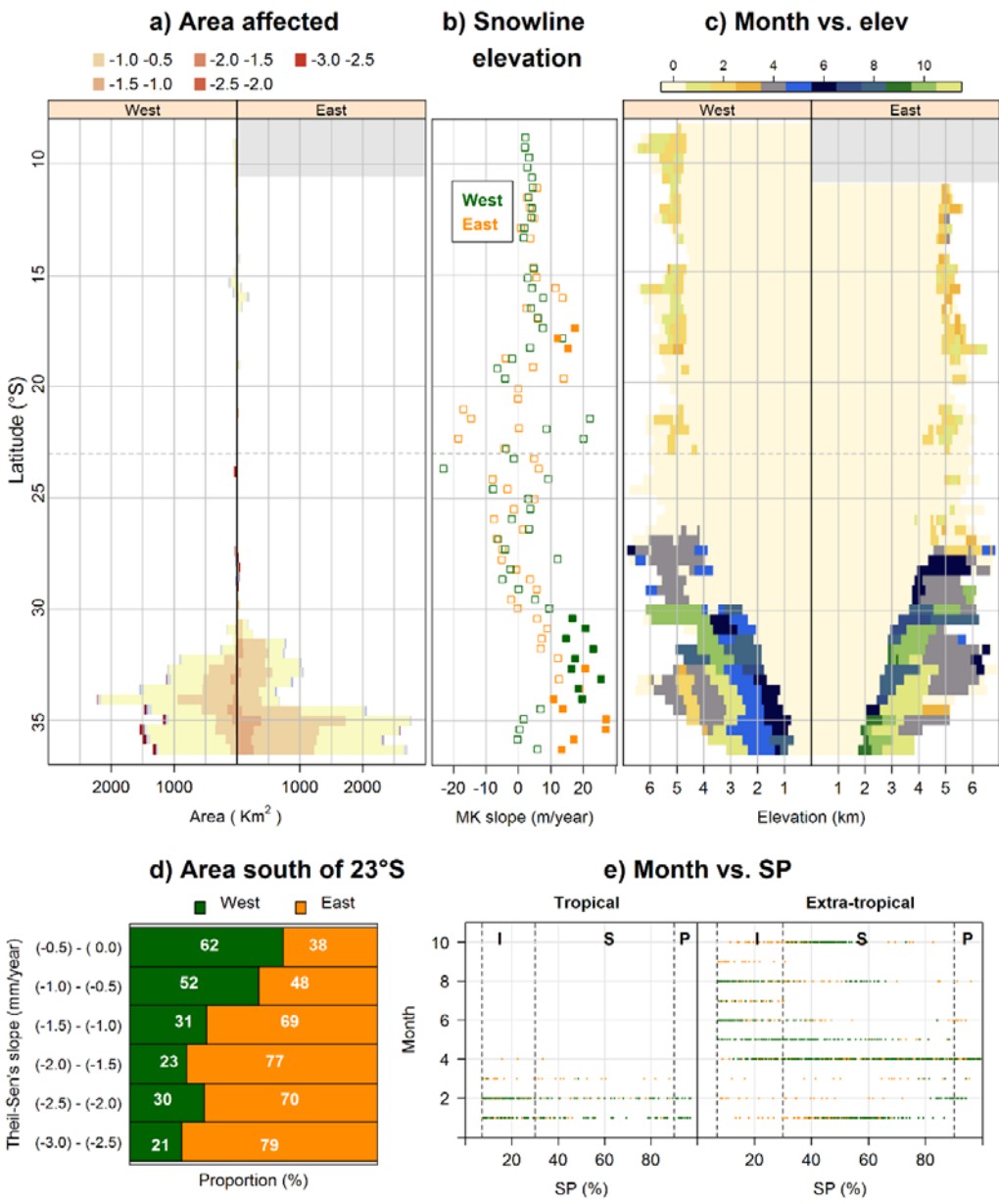

Figure 5. a) Areas with significant trends in annual SP by side of the continental divide (west/east) and latitude band, b) rate of change (Theil-Sen's slope) in the elevation of the snow line (SP=20%); solid symbols have significant trends (p≤0.05), and c) months with the highest significant Theil-Sen slope from the Mann-Kendall trend analysis of monthly SP; d) Percent of area in each Theil-Sen slope category for West and East sides, and e) months with greatest SP rate of change (slope) in tropical (North of 23°_S) and extra-tropical (South of 23°_S) latitudes for west (green) and orange (east) sides of the continental divide. Vertical dashed gray lines in (e) represent thresholds for intermittent (I), seasonal (S), and persistent (P) snow zones.

### 3.2 Climate connection

Many portions of the study area had significant trends in both annual precipitation and mean annual temperature from 2000-2016 (Figure 6). Trends in annual precipitation were primarily not significant north of 20°_S, whereas they showed a

5   significant decrease from 2-4 mm year$^{-1}$ at 28°_S to 15-25 mm year$^{-1}$ at 30-36 °C (Figure 6a) on the west side of the continental divide. The standardized decrease of precipitation (Theil-Sen's slope/mean annual P; Figure 6b) was low north of 31°_S (1%) and increased to the south, with higher values on the west side of the range. This indicates greatest impacts of decreasing precipitation on the west side. The seasonality of decreasing P (Figure 6c) shows that the austral winter (JJA) was the most affected season on the west side, and early austral fall (March) was most affected on the east side.

10  Temperature increased significantly north of 20°_S on both sides of the Andes in the range of 0.04 to 0.08 °C year$^{-1}$ (Figure 6d). In most of the rest of the study area, temperatures had increasing trends or no trends detected. A large area with significant increasing trends extended from 31-36°_S on the west side and 26-36°_S on the east side. The standardized change of temperature (Theil-Sen's slope / mean annual temperature) shows highest values north of 20°_S (5-15%) and a lower rate of increase (0.2-5%) from 31-36°_S (Figure 6e). The seasonality of increasing temperature trends varied across the region

15  (Figure 6f). North of 15°_S the highest increase occurred in June, and south of 26º_S, the greatest increases were mainly in fall months.

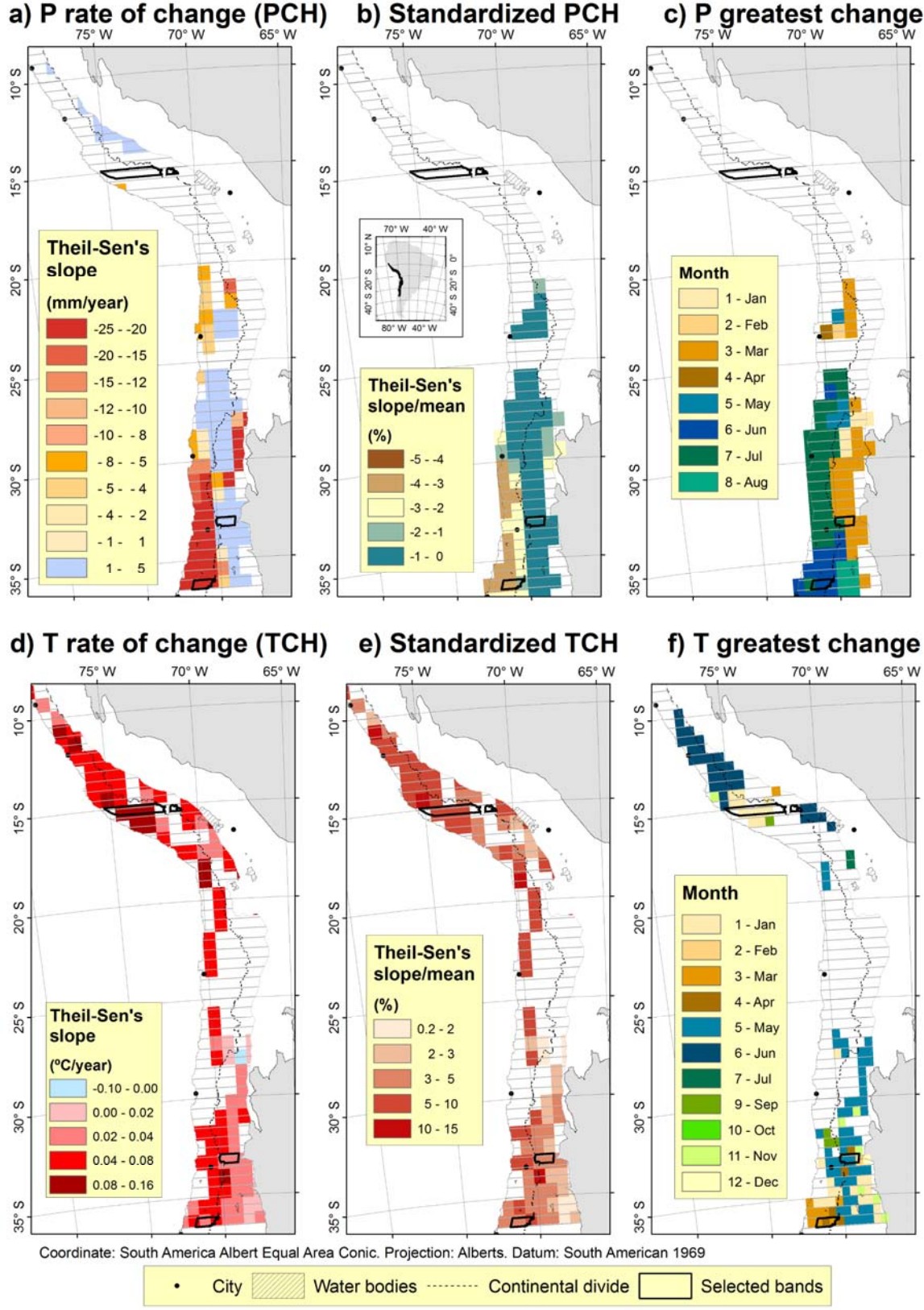

Figure 6. Theil-Sen's rate of change over the period 2000 through 2016, standardized Theil-Sen slope, and month of the greatest change (largest increase of temperature or largest decrease of precipitation) at an annual time scale for precipitation (P) (a, b, and c plots) and air temperature (T) (d, e, and f plots); only pixels with significant trends (p≤0.05) are colored. Black polygons highlight selected latitude bands used to describe in detail the evolution of temperature, precipitation, and snow from 1960-2016 (Figure 7).

Examples of climate variability across the region are illustrated in latitude bands 15°, 33°, and 36°_S in Figure 7. The latitudinal bands selected (Figure 7c-e) show a significant increase of temperature in both periods, 1960-2016 and 2000-2016, except in the tropical band (Figure 7c) where the increase was not significant. Precipitation had a non-significant increase for the northern bands (Figure 7c-d) and a non-significant decrease in the southern band (Figure 7e) for the period 1960-2016. After 2000, all bands showed a decrease of P, but this was only significant in the southern band (Figure 7e). SP showed decreases in all bands, but just the middle band was significant (Figure 7d).

ENSO (Figure 7a), PDO, and SAM indices (Figure 7b) show a cyclical pattern between 1960 and 2016. ONI is the primary indicator from NOAA for monitoring El Niño and La Niña events. NOAA categorizes an El Niño event when five consecutive 3-month running mean values of ONI are +0.5 or higher (warm phase) and a La Niña event when this value is -0.5 or lower (cold phase). Based on ONI values, very strong Niño events were defined in 1982-83 and 1997-98, and strong Niño in 1965-66 and 1972-73. Strong Niña events were defined in 1973-74, 1975-76, and 1988-89 (colored column between Figure 7a and b).

All SST indices (Niño 3, 4, 1+2, and 3.4) and MEI generally have similar patterns to the ONI index, with negatives values in La Niña events and positive in El Niño events (red lines in Figure 7a). Prolonged periods of negative SOI values coincide with El Niño episodes, and positive values coincide with La Niña episodes events. TNI high values relate to the very strong El Niño events (82-83 and 97-98), but negatives values were not related with La Niña episodes (73-74, 75-76, and 88-89). Additionally, long periods of negative TNI (89-90 to 97-98 and 00-01 to 07-08) were not related with La Niña events defined with the previous indices. The PDO, which describes a different climate pattern from ENSO, showed a longer cycle than ENSO indices. Some positive values were associated with very strong El Niño events (82-83 and 97-98), but other high positive values were not related to El Niño events (92-93) (Figure 7b). SAM showed a long cycle and presented negative values in La Niña events (73-74 and 75-76) and positive values with El Niño events (82-83 and 97-98) (Figure 7b).

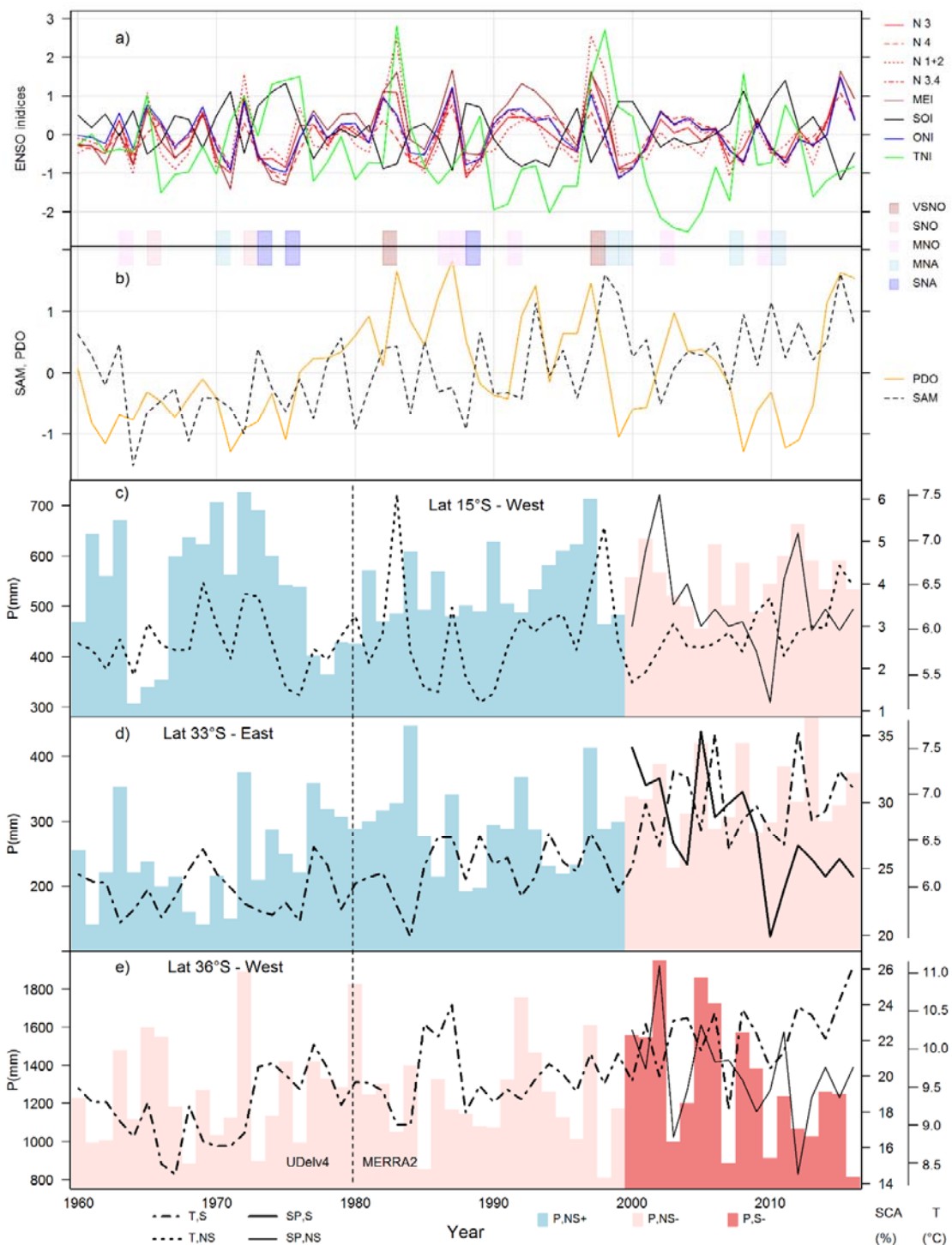

Figure 7. Comparison of the study time period (2000-2016) to the longer period of record since 1960. a) time series of ENSO indices of Niño event (Niño 3 (N 3), Niño 4 (N 4), Niño 1+2 (N 1+2), Niño 3.4 (N 3.4), Multivariate ENSO index (MEI), Southern Oscillation Index (SOI), Oceanic Niño Index (ONI), and Trans-Niño Index (TNI)), (b) Pacific Decadal Oscillation (PDO), and Southern Annular Mode (SAM). Colored columns between a) and b) represent Niño events based on the ONI index as Very Strong Niño (VSNO), Strong Niño (SNO), Moderate Niño (MNO), Moderate Niña (MNA), and Strong Niña (SNA). (c to e) plots show selected latitude bands (Figure 6). Time series of temperature show a non-significant increase (dashed line, (T,NS)) and significant increase (point-lined line, (T,S)) from UDelv4 and MERRA2 datasets. Precipitation (solid column) has non-significant increase in light blue, (P,NS+) (pink, (P,NS-)) and significant trend in bold colors (P,S-) from UDelv4 and MERRA2 datasets. Annual average SCA from MODIS dataset as percentage of whole latitude band 2000-2016 is represented where significant as a thick line (SP,S-) and a thin line where not significant (SP,NS+). The vertical dashed line in c to e plots shows the time change from UDelv4 to MERRA2 datasets.

Correlations between annual snow persistence and air temperature, precipitation, and the best climate index by snow region vary across the length of the Andes (Figure 8). The climate index selected to correlate with each snow region is the index and month with the greatest integrated correlation across the region (supplementary material 2). The highest integrated correlations per region were El Niño 3.4 centered in March (FMA) for north of 12° S (Tropical 1); El Niño 1+2 in January (DJF) for 2 - 24° S (Tropical 2); El Niño 1+2 centered in June (MJJ) for 24 -31° S (mid-latitude 1); and SAM centered in January (DJF) for areas south of 31° S (mid-latitude 2). Table 1 presents a summary of the best indices by snow regions.

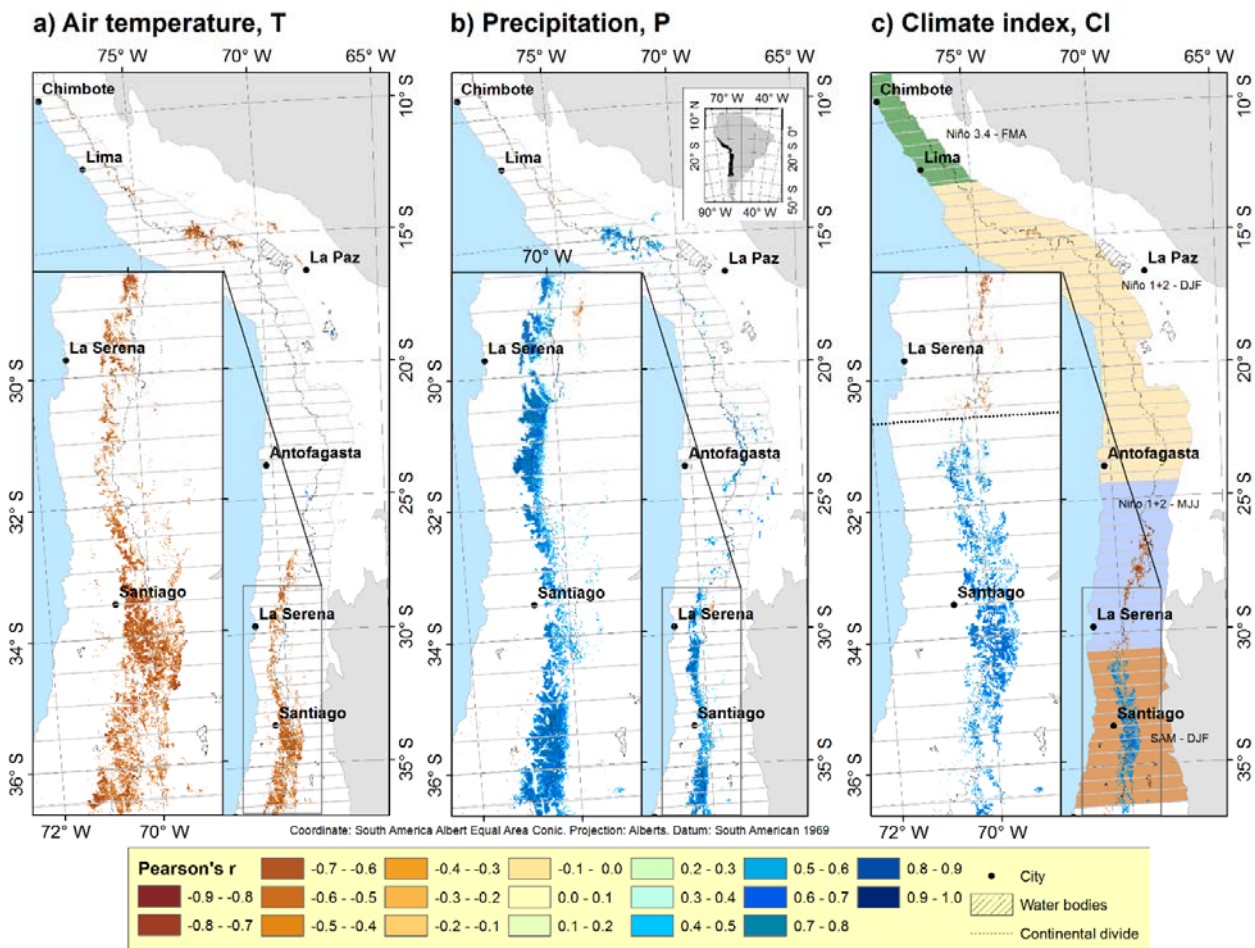

Figure 8. Map of Pearson's correlation coefficient (*r*) between 2000-2016 annual SP and a) mean annual temperature, b) annual total precipitation, and c) the best index/month climate index for SP prediction in each region.

Table 1. Top ranking climate indices for predicting SP by snow region indicating the average correlation ($r_{avg}$) and the sum of the absolute values of *r* for each snow region ($\sum|r|$).

| Snow region | Latitude range | Ranking | Index | Month | Area (km$^2$) | $r_{avg}$ | $\sum|r|$ |
|---|---|---|---|---|---|---|---|
| | | 1 | Niño 3.4 | FMA | 275 | -0.63 | 711 |
| Tropical 1 | 9-13° S | 2 | Niño 4 | DJF | 254 | -0.61 | 653 |
| | | 3 | ONI | DJF | 248 | -0.61 | 639 |
| | | 1 | Niño 1+2 | DJF | 1,277 | -0.60 | 3,122 |
| Tropical 2 | 13-24° S | 2 | Niño 3 | DJF | 408 | -0.57 | 945 |
| | | 3 | Niño 1+2 | FMA | 287 | -0.44 | 829 |
| | | 1 | Niño 1+2 | MJJ | 6,160 | -0.61 | 15,272 |
| Mid latitude 1 | 24-31° S | 2 | Niño 1+2 | JJA | 3,863 | -0.59 | 9,171 |
| | | 3 | Niño 1+2 | JAS | 3,426 | -0.59 | 8,161 |
| | | 1 | SAM | DJF | 17,790 | 0.59 | 42,511 |
| Mid latitude 2 | 31-36° S | 2 | SOI | AMJ | 16,402 | -0.60 | 39,480 |
| | | 3 | Niño 1+2 | DJF | 5,465 | -0.56 | 12,319 |

Air temperature, precipitation, and climate index were correlated with SP in some portions of the study area; however, patterns and relationships varied regionally. Both temperature and precipitation were significantly correlated with SP at around 15° S and south of 28° S (Figure 8a, b). For air temperature the relationship with SP was inverse except in a few high elevations (>5000 m) (Figure 9a), and the strength of the correlation was greatest in the intermittent snow zone (Figure 9d). Precipitation had a direct correlation with SP in almost all places where the correlation was significant (Figure 9b). The strength of this correlation varied with SP in extra-tropical areas (Figure 9e), with the strongest correlations occurring in the seasonal snow zone. The correlations of SP with climate indices were mostly negative north of 31° S, where ENSO indices were used, and positive south of this latitude, where SAM was used as the climate index (Figure 8c, 9c). These correlations did not have a clear pattern with change of SP in the tropical latitudes but were highest in the seasonal snow zone in extra-tropical latitudes (Figure 9f).

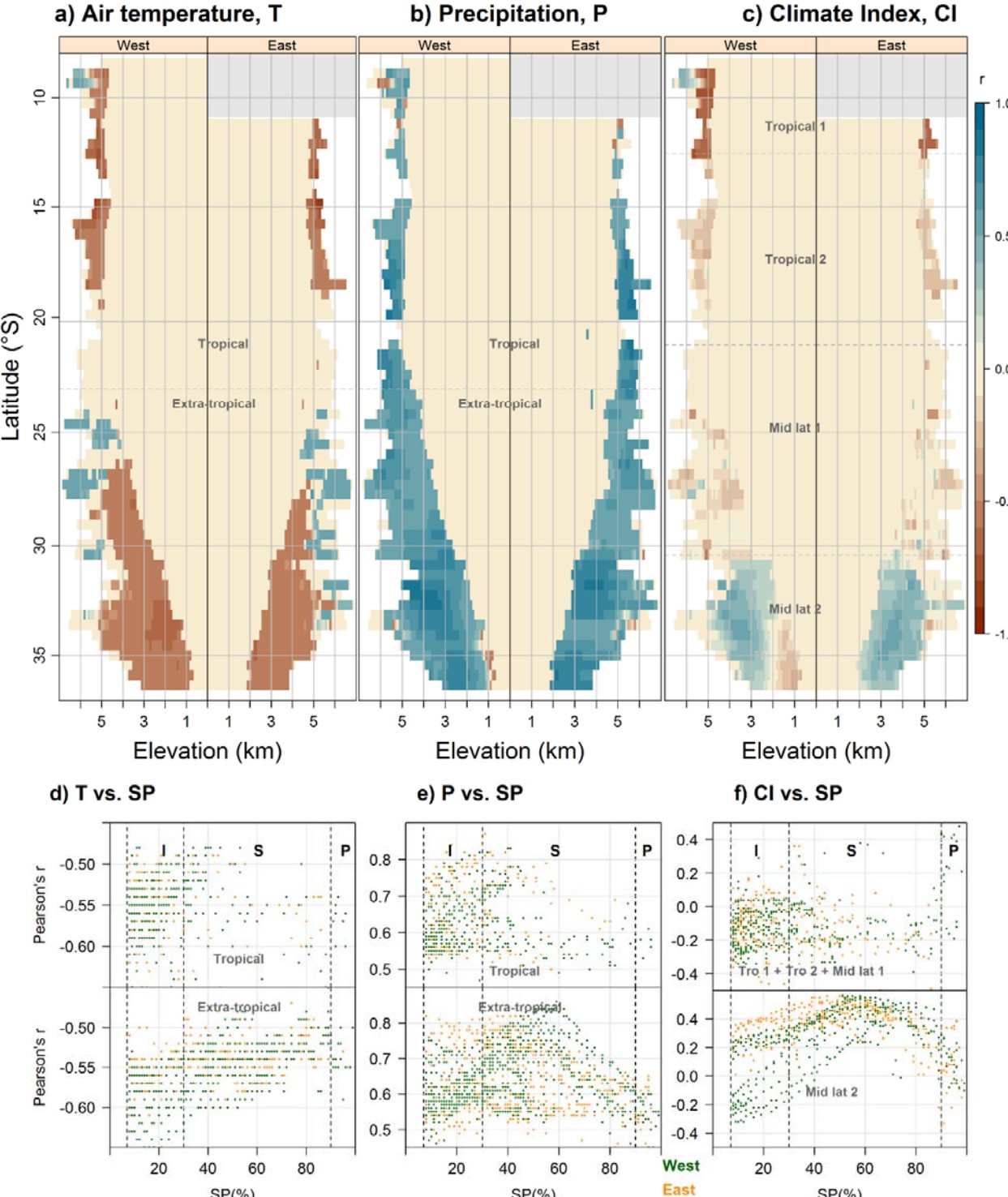

Figure 9. Latitude-elevation band analysis by side of the continental divide (west/east) for Pearson's correlation coefficient (*r*) between annual snow persistence and a) air temperature, b) precipitation, and c) climate indices. Grey areas represent latitudes masked due to high frequency of cloud cover in snow covered area analysis (>30% period). Plots d), e), and f) show values for each latitude-elevation band change by SP in West (green) and East (orange) sides. Vertical dashed grey lines represent thresholds for intermittent (I), seasonal (S), and persistent (P) snow zones.

To examine the combined influences of precipitation and temperature on annual snow persistence, we ran a coefficient of determination ($r^2$) analysis using P and T as predictors (Figure 10a). We excluded climate index from this analysis because of its correlation with P and T. Areas south of 28°_S had significant and strong coefficient of determination ($r^2$) values (27,530 km$^2$). Around 75% of the area with significant $r^2$ had values between 0.4-0.7. To explore which parameter (air temperature or precipitation) most influenced SP in each location, we computed the relative importance of each variable in multiple linear regressions (Figure 10b). Precipitation was the most important predictor across much of the region except from around 33-34.5°_S and at the lower elevations of each affected region

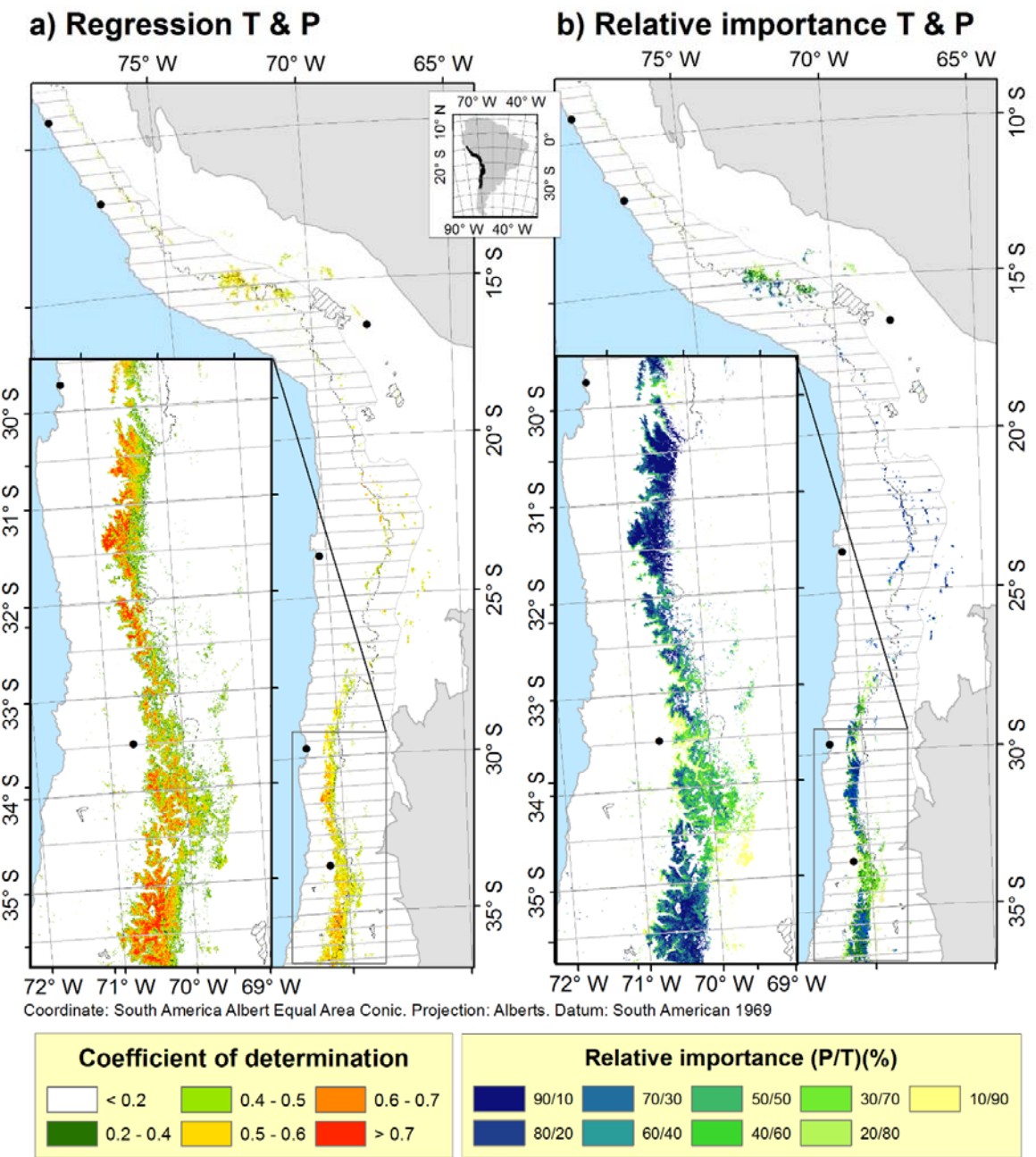

Figure 10. Multiple linear regression analysis by pixel showing a) the coefficient of determination ($r^2$) for SP predictions using annual precipitation and mean annual air temperature) and b) Relative importance (RI), indicating the proportion of (P/T) individual contribution of precipitation to the full $r^2$ of the model.

High values of $r^2$ north of 25°S are found at high elevations (> 5000 m), and south of this latitude high $r^2$ values are found at lower elevations and show an elevation dependence in the Middle Latitude 2 region (Figure 11a). The maximum $r^2$ values in the Middle latitude 2 region are generally in the seasonal snow zone on the west side (Figure 11c). The relative importance plot shows a clear importance of T in tropical areas (north of 15° S) and at lower elevations south of 26° S on the west side. The highest relative importance of P was in the seasonal snow zones, whereas the relative importance of T increased in intermittent and persistent snow zones (Figure 11d).

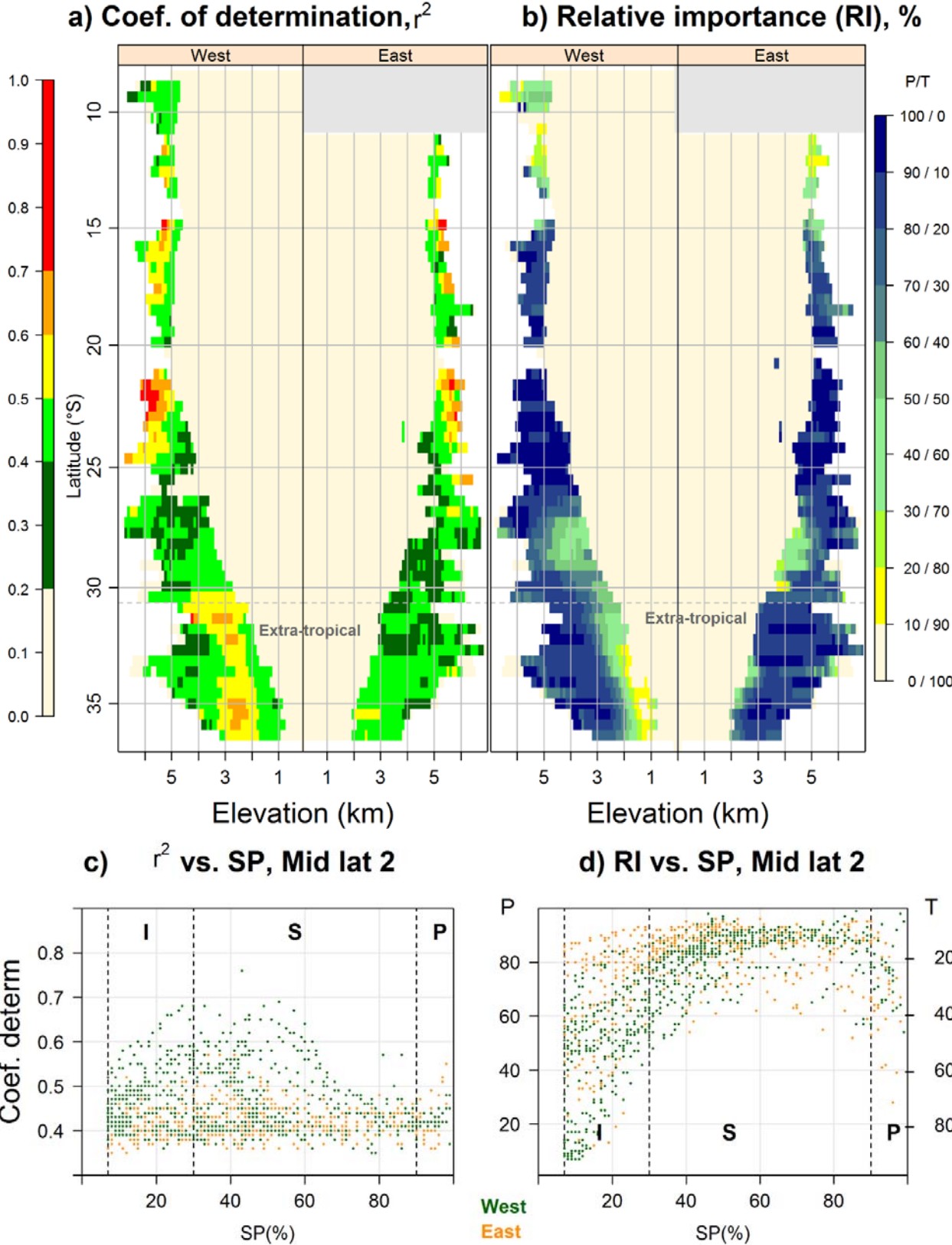

Figure 11. Latitude-elevation band analysis by side of the continental divide (west/east) for a) coefficient of determination ($r^2$) for multiple linear regression using annual precipitation and mean air temperature to predict SP, 2000-2016, b) relative importance (RI) of P/T. Grey areas represent latitudes masked due to high frequency of cloud cover in snow covered area analysis (>30% period). c) $r^2$ vs. SP and d) RI of P/T in Mid Latitude 2 snow region for West (green) and East (orange) vs SP values. Vertical dashed grey lines inside inset boxes represent thresholds for intermittent (I), seasonal (S), and persistent (P) snow zones.

**4 Discussion**

The range of spatial and temporal snow patterns during the period analyzed provide important insights about snow patterns and how they relate to temperature and precipitation during the period of the analysis. However, we note that the main limitation of our study is the short time period of the study (2000-2016), so the trends presented should not be considered representative of long-term climate trends (> 30 years). This study provides a basis for future studies testing longer-term relationships of spatiotemporal tends in snowpack with broad-scale climate drivers, and with weather and climate variability in the region.

4.1 Spatial variability in snow persistence trends

Snow persistence trends varied across the study area. North of 25° S, the snow-covered areas are small, making it difficult to track trends in SP (Figure 3a). This low snow presence is likely due to a combination of temperature and precipitation effects. Precipitation is concentrated during the austral summer, synchronous with the highest temperatures of the year (Figures 2a and b). Thus, precipitation falls mostly as rain, and snow is limited to elevations >5000 m (Figure 4a), where temperature is generally cold. The limited trend is SP in this area (Figures 3b and 4b) is likely because SP variability is more strongly related to precipitation than to temperature (Figure 11b), and few trends in precipitation were detected in those areas (Figure 6a).

South of 25° S, we detected a significant decrease of SP (Figure 3b). The rates of decline varied across the range of elevation and latitudes. Areas with intermittent winter snow showed a moderate decrease (-0.5 to -1.0% per year) (Figure 4e), with the largest decrease during winter (Figure 5c). However, these areas had the largest relative rates of snow loss (standardized slope in SP) (Figure 4c,f), particularly on the east side of the Andes. Low-elevation snowpack is often found to be most sensitive to climatic change, yet we found that areas with seasonal winter snow showed the steepest absolute rate of change (-1.5% per year) around SP=60% (Figure 4e). Elevation variability in snow trends was also present in the San Francisco estuary and its upstream watershed (California, USA), where the greatest loss of snowpack was identified in the 1300–2700 m elevation range (Knowles and Cayan, 2004) in the seasonal snow zone (Moore et al., 2015). This spatial variability in snow trends with elevation, latitude, and side of the mountain range demonstrates the importance of spatial information for documenting snow change; trends identified at specific field sites may not be representative of the region as a whole (Fassnacht et al., 2016).

Loss of snow in the intermittent snow zone affected the trend of snow line elevation. Increases in snow line elevation have been documented in Andes tropical areas for the 1961–2012 time period (Pepin et al., 2015). However, our work does not show an increase in snowline in tropical areas, probably due to the short period of record, precipitation dependence of snow cover in this area (Saavedra et al., 2017), and the lack of consistent precipitation trends in these latitudes over the 2000-2016 period (Figure 6a). In the central area (32-34° S), Carrasco et al. (2008) found an increase of elevation of isotherm 0 °C of 23 m year$^{-1}$ that is consistent in both magnitude and trend direction with our results (Figure 5b), probably due to the greater temperature-dependence of snow line in this area.

Previous studies had shown inconsistent findings related to snow changes for the same latitude range where we detected declining snow persistence (30–37° S). Several studies have documented increasing snow in the region. For example, Masiokas et al. (2006) showed a positive trend (not significant) of annual maximum snow water equivalent at five ground stations between years 1951–2005. These results differ from our study in the same area, probably because (1) the limited in situ observations that they used may not have been representative of the region as a whole, (2) the longer time period covered (54 years) may have had a different trend than the 2000-2016 time period examined here (Venable et al., 2012), and

(3) the maximum snow water equivalent may not have the same trends as SP in this region. Another study found increases in snow days but decreases in maximum snow cover extent based on SnowModel simulations (1979–2014) in the Andes Cordillera south of 23° S (Mernild et al., 2017). Their findings highlight how interpretation of snow trends may vary with the snow variable selected. Again, the time period of analysis was different from our study, and this would also contribute to

differences in snow cover trends, as trends identified in a time series can vary both in magnitude and direction for different time intervals (Fassnacht and Hultstrand, 2015).

Many prior studies have found declines in snow that are consistent with our findings here. Prieto et al. (2001) documented a reduction in days with snow between years 1885-2000 in the Mendoza area of Argentina (32.5° S, elevation 750 m) based on newspaper weather reports, and this trend is consistent with our results in this area. Studies focused on glaciers found a

general decrease in areas of snow/ice during time periods from 1983-2011 (Cortés et al., 2014), 1955-1997 (Pellicciotti et al., 2013), as well as over the last 100 years (Masiokas et al., 2009). Glacier-focused studies do not capture the geographic range of our snow cover analysis, but these declines in glacier area are consistent with our findings of declining snow persistence

### 4.1 Climatic causes of snow persistence trends

Temperature increased throughout most of the Andes region, whereas SP only declined significantly in mid-latitudes during the 2000-2016 time period. Many studies of snow pack change have focused on the importance of temperature to snow trends. In a warmer world, less winter precipitation falls as snow, and the melting of winter snow occurs earlier in spring (Barnett et al., 2005), explaining why snow cover is declining in most parts of the world when considered over large areas (Brown and Robinson, 2011). Yet site-specific snow trends vary (Vaughan et al., 2013; Fassnacht et al., 2016), likely due to

the combined roles of temperature and precipitation affecting the snowpack (Adam et al., 2009; Stewart, 2009). Indeed, we found that trends in SP were detectable only where temperature was increasing while precipitation was decreasing (Figure 6). In our study, precipitation emerged as the most important variable affecting SP across the mid-latitude Andes, but temperature also affected SP in warmer areas such as the tropical latitudes and lower elevations of snowpack in mid-latitudes. Climate modelling studies suggest that the trend of increasing temperature and declining precipitation will continue

in this area (Bradley, 2004; GCOS, 2003), which will in turn affect streamflow in the region, where many of the rivers have snowmelt-dominated runoff (Cortés et al., 2011).

Both T and P are modulated by ENSO in the region (Masiokas et al., 2006; Santos, 2006; Zamboni et al., 2011; Meza, 2013; Valdés-Pineda et al., 2015b). In the tropics, El Niño conditions relate to low precipitation and warmer air temperatures, whereas in higher latitudes El Niño relates to higher precipitation and warmer temperatures (Garreaud et al., 2009). Our

results suggest that this ENSO influence on temperature and precipitation most affects snow patterns north of 30° S. The trends of decreasing P and increasing T we found in these latitudes are generally consistent with previous studies (Vuille and Bradley, 2000; Bradley, 2004; Quintana, 2012; Salzmann et al., 2013; Kluver and Leathers, 2015). South of 35° S, SAM is the climate index best correlated with SP. The influence of SAM on South American climate has been documented in other studies (Vera and Silvestri, 2009; Fogt et al., 2010), and it has been found to be an important control on radial tree growth

(ring width) in the same latitude range where we identified its influence on snow (Villalba et al., 2012). These relations between SAM and both tree rings and snowpack may indicate that SAM is influencing tree growth through its influence on snowpack characteristics. More specifically, snowpack can influence growing season length and water availability during the growing season. Given the relatively dry summers of this region, increased snowpack would be expected to increase radial tree growth by providing a source of moisture during the growing season. We present the first study relating SAM to snow

response in South America, and these relations suggest that tree rings could be a viable method to reconstruct snowpack in

the southern Andes (Woodhouse, 2003). However, we note that the short duration of our study time period is insufficient for complete analyses of how snow patterns relate to ENSO and SAM. During the period of our study SAM was mainly positive (Figure 7b), so a long-term study is required to determine whether different phases of SAM have different snow cover responses.

The influences of climatic indices on snow are not always consistent with their influences on precipitation and temperature because the timing of precipitation and temperature anomalies is particularly important for snow. We found that in intermittent snow zones, where snow is not consistently present throughout the year, the greatest decreases in SP were during austral winter. Here, a decline in winter precipitation leads to a decline in snow and an increase in the elevation of the snow line. Higher up in the seasonal snow zone, where snow is present every year, the negative trends in SP were strongest in
spring. These trends are explained primarily by temperature increases that accelerate the spring loss of snow, but they are also affected by the amount of winter and spring precipitation adding to snow accumulation.

Results of this study are affected by the accuracy of the snow cover and climate data. For snow cover we used the 8-day product to minimize cloud impairment, but this limited temporal resolution to 8 days. Future testing could evaluate whether these trends are consistent using daily or fractional SCA products (Rittger et al., 2013). These finer temporal resolution
products will likely need to incorporate additional cloud removal algorithms (Gafurov and Bardossy, 2009; Gao et al., 2011; Hall et al., 2010) in many areas to make an analysis feasible. Future analyses could therefore also examine how cloud presence may affect the direction and magnitude of snow persistence trends. During the development of this paper, we tested several different gridded climate data products UDelv4, MERRA2, ERA-Interim (Berrisford et al., 2011), and Global Historical Climatology Network-Monthly (GHCN-M) (Lawrimore et al.,2011). Generally these products produced similar
conclusions about how they affect SP; however, there were notable differences in both the patterns and magnitudes of precipitation and temperature in these products. For example, there were no significant trends in precipitation north of 20° S in UDelv4, whereas some trends were present in MERRA2 (Figure 6). UDelv4 showed some decreasing temperature trends from 15-29° S, whereas MERRA2 had only increasing temperature trends in this area (Figure 6). These examples highlight the importance of improving the quality of gridded climate products in the Andes.

## 5 Conclusions

This work quantifies trends in snow persistence across a large range of latitude (9-36° S) and elevation (0-6961 m) on both sides of the Andes Mountains from 2000-2016. North of 29° S (tropical latitudes and desert Andes), minimal changes in SP were detected because the areas with snow are small, and there were limited changes in precipitation from 2000-2016. South
of 29° S, significant loss of SP was observed in association with both increased temperature and decreased precipitation. Sixty-two percent of the area with significant SP loss was on the east side of the Andes Mountains. The absolute magnitude of these losses was greatest in areas with seasonal winter snow, whereas the greatest relative loss of snow persistence was in areas with intermittent winter snow, where the snow line increased in elevation. The relative importance of precipitation and temperature to SP varied with latitude and elevation. In most of the study region, precipitation had the greatest relative
importance for SP, but temperature increased in importance in tropical latitudes and at lower elevations of mid-latitudes. SP was highly correlated with temperature, precipitation, and climate indices across the region, with ENSO indices better predicting SP north of 31° S and the SAM better predicting SP south of 31° S. The time period analyzed in this study is too short to document long-term snow changes, but they highlight the latitude and elevation variability in snow trends and the combined importance of precipitation and temperature in snow persistence. Continued monitoring of the spatial patterns of
snow trends will help identify areas that are highly sensitive to climate change and aid in future water supply planning.

## Acknowledgments, Samples, and Data

Saavedra's research has been supported by the Chilean government through the Fulbright-CONICYT scholarship program with additional support from NSF EAR-1446870 and CONICYT-FONDECYT Posdoctorado 3170651.

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
