# Peer review of "Changes in Andes Mountains snow cover from MODIS data 2000-2016"

_The Cryosphere, 2017_

## Referee Comment (RC1) · Anonymous Referee #1 · 3 Jun 2017

The paper presents an analysis of snow cover persistence trends over the Andes Mountains based on MODIS data for the period 2000 – 2014. The trends are related to various meteorological parameters (air temperature and precipitation) and large-scale oceanic–atmospheric indices (e.g. ENSO, PDO, SAM), and are distinguished among different latitude ranges and eastern vs. western sides of the Andes Ranges. The rationale provided for the study is that snow cover area variability and trends have not been studied in detail for South America and that there is a need to document these changes.

While this seems to be reason enough to carry out a remote sensing study on changes in snow cover for this vast region, the authors do little to provide much context on the importance of the study, and I struggle to understand the value or the contribution to

the scientific literature that it makes. Of most concern is the extremely limited length of the observational dataset (only over one decade) upon which the trends are based. This is a fundamental flaw that seems to have been overlooked entirely – indeed there is no mention of the limitations of the data in this regard anywhere in the paper. Trends cannot be reliably detected and distinguished from short-term or periodic variations over such a short period, and the observed changes certainly cannot be correlated to long-term trends in meteorology or patterns of atmospheric–oceanic variation. Thus all subsequent interpretation of the results is rather meaningless. Perhaps there could have been more effort to place the short term snow cover patterns into some context by linking these to available longer-term surface observations, which could in turn be correlated with the other variables.

Unfortunately I do not see a major contribution here that warrants publication in The Cryosphere and must recommend rejection. The advice I have would be to seek to expand the analysis to include other available data sources that can better help shed insight on the longer term trends and variability, but this would beyond the scope of a major revision.

---

## Referee Comment (RC2) · Anonymous Referee #2 · 12 Jun 2017

I agree with the first referee that the length of the study period (14 y) is short to analyze trends in the snow cover, especially in the Andes where the climate has a strong interannual variability due to the influence of ENSO. However, the authors did test the significance of the trends hence I do not think that it is a methodological flaw, but rather a strong limitation of the study that would indeed deserve a better discussion.

My main concern is the lack of literature review in the introduction. This was also noted by the first referee but in my opinion this is the main issue with this paper. Some relevant studies are cited in the Discussion, but the authors should use them in the introduction to better establish the current knowledge gaps. There are not so many papers about the snow cover variability in the Andes (especially at this scale). Therefore, the scope of the literature review could be extended to the climate, hydrology,

glaciology studies. A fair review of the existing knowledge is necessary to identify knowledge gaps beyond this unjustified statement "The impacts of climate change on snow covered areas in South America have not been studied in detail due to sparse and unevenly distributed climate data" and would help define the contribution of this paper.

I see also an issue with the MODIS snow cover data. The authors used the MOD10A2 v5 snow product "to minimize cloud impairment". However there remain gaps due to cloud obstruction in this product, which is a composite over an 8-day period. I wonder if even a relatively low number of cloud pixels could generate spurious trends, in particular in this region, where the cloud cover can be correlated with ENSO (cloud cover seasonal distribution is not random). Many gap filling techniques exist and could be used to generate a cloud-free dataset before performing trend analyses.

---

## Author Response (AR1)

*We thank the reviewer #1 for her/his critical comments that help us to improve our research.*

**Reviewer:** The rationale provided for the study is that snow cover area variability and trends have not been studied in detail for South America and that there is a need to document these changes. While this seems to be reason enough to carry out a remote sensing study on changes in snow cover for this vast region, the authors do little to provide much context on the importance of the study, and I struggle to understand the value or the contribution to the scientific literature that it makes.

*Response: We have strengthened the description of this contribution in the revision. While snow trends have been studied over small segments of this region, this is the first comprehensive snow trend study that covers the region as a whole using observations alone rather than modeled reconstructions. We argue that observation-based studies are a useful complement to model-based studies, which are affected by uncertainties in input data. Moreover, multiple lines of evidence are always valuable in trying to identify spatiotemporal trends and drivers of processes such as snow.*

*The snowpack provides a large reservoir of water in the region where Peru, Bolivia, Chile, and Argentina all depend on snow and/or glacier melt for water supply. Additionally, seasonal snow is a critical component of the surface energy balance and hydrologic cycle, particularly at high elevations and high latitudes. We included all these arguments in a new version of our manuscript.*

**Reviewer**: Of most concern is the extremely limited length of the observational dataset (only over one decade) upon which the trends are based. This is a fundamental flaw that seems to have been overlooked entirely – indeed there is no mention of the limitations of the data in this regard anywhere in the paper. Trends cannot be reliably detected and distinguished from short-term or periodic variations over such a short period, and the observed changes certainly cannot be correlated to long-term trends in meteorology or patterns of atmospheric–oceanic variation. Thus all subsequent interpretation of the results is rather meaningless.

*Response: We agree that this is the most limiting factor of our analysis, but it is not a methodology error. Our manuscript addresses trend interpretation in detail by relating snow trends to corresponding temperature, precipitation, and climate cycles. We did not claim in the previous version of the manuscript that these trends are correlated with long-term trends, and we have added additional text in the discussion to make this point in greater detail. Trend analyses based on the time period of the MODIS record are common in the literature. Related to snow and ice patterns, high profile examples are Hall et al.*

*(2013) Geophysical Research Letters and Šmejkalová et al. (2016) Nature Scientific Reports. Researchers choose to use the MODIS record alone in some cases because other products are not comparable in resolution, time step, and/or spatial coverage. This study also provides a basis for future studies testing longer-term relationships of spatiotemporal tends in snowpack with broad-scale climate divers, and weather and climate variability and change in the region.*

*We appreciate the suggestion to link this analysis to longer-term surface observations and agree that this is important for context of the study. We refer the reviewer to Figure 7, where this was done in the original version of the manuscript. Our analysis of how snow persistence relates to climate variables is also a means of relating the snow patterns observed since 2000 to other climate records, which are generally available for longer time periods.*

**Anonymous Referee #2**

*We thank the reviewer #2 for the constructive comments and suggestions.*

**Reviewer**: My main concern is the lack of literature review in the introduction. This was also noted by the first referee but in my opinion this is the main issue with this paper. Some relevant studies are cited in the Discussion, but the authors should use them in the introduction to better establish the current knowledge gaps. There are not so many papers about the snow cover variability in the Andes (especially at this scale). Therefore, the scope of the literature review could be extended to the climate, hydrology, glaciology studies. A fair review of the existing knowledge is necessary to identify knowledge gaps beyond this unjustified statement "The impacts of climate change on snow covered areas in South America have not been studied in detail due to sparse and unevenly distributed climate data" and would help define the contribution of this paper.

**Response**: *Literature review: We extended the introduction to include more relevant research and better illustrate the gap of information filled by our study.*

**Reviewer**: I see also an issue with the MODIS snow cover data. The authors used the MOD10A2 v5 snow product "to minimize cloud impairment". However there remain gaps due to cloud obstruction in this product, which is a composite over an 8-day period. I wonder if even a relatively low number of cloud pixels could generate spurious trends, in particular in this region, where the cloud cover can be correlated with ENSO (cloud cover seasonal distribution is not random).

**Response**: *Cloud effects on trends: We tested the trends of cloud cover during the same period to evaluated whether these could affect snow cover trends. We did not detect trends in cloud cover in the study area during the 2000-2014 period, which is one indication that cloud impairment may not have affected the trend directions. We included this information in the new version of manuscript.*

**Reviewer**: Many gap filling techniques exist and could be used to generate a cloud-free dataset before performing trend analyses.

*Response*: *We have also explored cloud correction approaches in detail for this area but have some concerns about generating a cloud-free dataset before performing trend analyses. The majority of MODIS gap filling techniques include data from Terra and Aqua satellites as the first step. Since band 6 in Aqua is defective, the performance of these algorithms must be tested before applying the correction. In the new collection 6 of MODIS snow data, the reconstruction of band 6 data on Aqua may produce some spatial-temporal changes in snow patterns that would be interesting to test in the region at a daily step, but we did not detect a problem with cloud impairment in the 8-day product used. We extended the discussion section to include cloud filling options.*

[revised manuscript text omitted]

---

## Editor Decision (ED1)

Dear. Dr. Saavedra,

Thank you for the revised manuscript. "Changes in Andes Mountains snow cover from MODIS data 2000-2016" (tc-2017-72). Following previous comments, you extended the study period, thank you. I respect your decisions regarding the use of Landsat and on the use of in situ data.

I have several suggestions for minor technical changes, but I do have one major concern that needs to be rectified. I am not sure but I think that it relates to the precipitation component of the ERA-Interim dataset. If you look at Figure 8b, you can see a very distinct artifact in the mapped data, especially in the inset. There is an "edge" that runs parallel to the line of longitude, and if you look closely at the main map, there are other short edges in the data that align with lines of latitude. The result is that there appears to be a distinct bias in the precipitation data. The main edge is roughly aligned with the continental divide. What are the effects of this artifact, especially on the subsequent latitude-elevation band analysis (Figure 9), and MLR analyses (Figure 10)?

If this error is introduced by the ERA-Interim dataset, then its use needs reconsideration. If the problem cannot be rectified, then it would seem that the manuscript may have to return to the shorter study period of the earlier version. In any case, this will take some time to resolve so I suggest minor revisions. Please take time to make the appropriate changes. The artifact is obvious, and I am surprised that it was not noticed, treated, or discussed in this revised version of the paper.

Sincerely,

Peter

Minor comments:

P1,L24: Delete "in magnitude"

P1, L34: Change "warming" to "increasing"

P2, L5: comma after (17°S to 31°S)
P2, L13: Change "related with" to "related to"
P2, L16 and elsewhere: There is always a space between the value the "°" symbol for temperatures
P3, L4-6: Change "The Andes Mountains cross seven countries (Vene … tina) along more than 8,000 km across a wide range of latitudes … They represent … and are the longest mountain chain…" to "The Andes Mountains cross seven countries (Vene … tina) and a wide range of latitudes … They represent … and are the longest (8,000 km) mountain chain…"
P3, Figure 1: Please put a black box in the main map around the area that is "zoomed in to" in the inset, and thicker lines connecting the two.
P4, L5: Comma after "area"
P4, L7: Change "In these latitudes" to "At these latitudes"

P5, Figure 2 (and other similar composed figures): It is not obvious that the West/East relates to continental divide. Perhaps you should include this in the caption. How about changing "Mean monthly hydroclimate variables by latitude…" to "Mean monthly hydroclimate variables by location with respect to the continental divide and latitude…"

P5, L16: Change "continuing in real time monthly" to "continues in real-time monthly increments"

P5, L18: Given the different resolutions of these datasets and the 0.5° working resolution of the paper, please include a line about how the data were combined and the interpolation of the ERA-Interim data. Was there any fitting of one to the other?

P6, L8: Change "surface temperature regions (SST)" to "surface temperature (SST)"regions"

P6, L15: Please define NOAA here and delete from P14, L8.

P6, L22: Change "America" to American"

P6, L38: R is used for all analysis, so move much of the text from P7, L10-11, to here. Change "Saavedra et al. (2017). For the remaining pixels we used the" to "Saavedra et al. (2017). For the remaining pixels we conducted geo-statistical analysis on the data using the statistical computing R software (R core Team, 2013). We used the"

P7, L14 and elsewhere: the parameters $r$ and $r^2$ should be italicized. This affects text throughout the main document, tables and figures.

P7, L26: Delete "show"

P8, L17: Comma after "29°S"

P10, L4: change ""common range of change values are in the range of" to "common change values are on the order of"

P13, Figure 6: The line symbol for the continental divide is not visible in the legend or the figure.

P15, L11: Change "is represented as a line for significant (not significant) in thick, (SP,S-), (thin, lines." To "is represented where significant as a thick line (SP,S-) and a thin line where not significant (SP,NS+)."

P15, L12: Change "of UDELv4 and ERA-Interim datasets" to "from UDELv4 to ERA-Interim datasets"

P15, Figure 7: Lines for N3 and MEI are indistinguishable. Traces are also very pixelated. I hope that final line art diagrams for this manuscript will be vector based and smooth. Classes of Niño/Niña events are difficult to distinguish and need more contrast.

P16, Figure 8: Obvious artifact in (b). Can't see the line symbol for the continental divide in the figure or legend

P19, Figure 10: Obvious artifact in (a) and (b).

P20, Figure 11: (b), is this PRI or RI? Please coordinate with caption. (c) part of figure title is missing. (d) is this RI or PRI?

P22, L16: Effect of apparent precipitation anomaly on results?

P23, L28: What is the effect of the apparent precipitation-related anomaly on this conclusion?

---

## Author Response (AR2)

We thank the reviewer #3 for the constructive comments and suggestions. Below we respond point-bypoint comments:

**1) Extend the study period to 2017 (17 years of data).**

Response: We extended the period to 2016 to cover 17 years of analysis (2000 to 2016). We did not include the year 2017 because (1) the annual cycle of snow for 2017 is not yet finished in the study area, and (2) the MOD10A2 Collection 5 data are only available until 2016. Note that adding the recent years required that we switch datasets for temperature and precipitation, as the dataset we used previously was not available through 2016.

**2) Consider the possibility of using Landsat data (or other sources) to increase the length of the snow cover dataset used. This point would significantly increase the impact of the paper and it would be the first long-term observational snow cover work for the Andes.**

Response: We agree that extending the length of analysis would add value to our results. However, our research is based on the frequency of snow, and for this purpose Landsat has two major limitations: (1) It has imagery every 16 days, and this probably cannot capture the snow trends in this region, and (2) it is more sensitive to cloud presence because the presence of cloud for one image date leads to month with no data (16+16 days). Additionally, the broad analysis we conducted with MODIS is first step to identify important locations to examine with finer spatial resolution data such as Landsat, Sentinel, etc.

Snow cover data from AVHRR could be used to extend the study further back in time, but these images have 1 km resolution, which further reduces the ability to detect spatial patterns in steep mountain regions. This could be done to extend our present work, but given the level of detail of analysis we conducted with MODIS data for this manuscript, we feel that adding a different sensor would detract from the paper's current focus.

Note also that the types of analyses we conducted were designed to link the more limited period of record from MODIS to longer time periods. The correlation analyses with precipitation and temperature and analyses of climate indices all help make this link (Figure 7).

**3) Furthermore, if observational in-situ data is considered for the 2000-2014 period then the conclusions of the paper would be more robust, especially in terms of the multiple linear regression analysis.**

Response: The main reason to develop a remote sensing approach for examining snow is the lack of field data in the study area. After a quality analysis from field stations, we identified less than 15 stations that could be used to compare to MODIS data. These stations are located across a distance of more than 2000 km and a wide latitude range and therefore are not useful for developing an elevation-snow frequency developed in our research. Given the discrepancies in scale between ground measurements and MODIS, we think a larger number of ground stations would be needed to make any strong conclusions about how the MODIS data compare with ground observations.

[revised manuscript text omitted]

---

## Editor Decision (ED2)

Dear. Dr. Saavedra,

Thank you for the revised manuscript. "Changes in Andes Mountains snow cover from MODIS data 2000-2016" (tc-2017-72). I respect your decisions regarding the use of MERRA2 data.

The manuscript is improved, but I have several suggestions for minor technical changes before you submit the manuscript for publication.

Sincerely,

Peter

Minor comments:

P1, L11: Here you say the Andes are 7,000 km in length, but on P3, L6, you say 8,000 km. I think most estimates are closer to 7,000 km.

P1, L18: Delete "Interim". Should ENSO, SAM, and PDO be spelled out, as you did for MODIS? Similarly, Should MERRA2 also be spelled out? I think so given our broad readership.

Throughout: As before, there is always a space between the value the "°" symbol for temperatures, but not for position (latitude or longitude). I think you may have carried out a blanket search-and-replace that introduced some errors.

Figure 6: The line symbol for the continental divide is still not clearly visible to me in the legend or the figure. Can you make the lines thicker?

P15, L11: Change "think" to "thick"

P22, L3: Delete "up front"
P24, L17: Change "For climate data, we tested several different gridded products" to "During the development of this paper, we tested several different gridded climate data products"

---

## Author Response (AR3)

We thank the editor Peter Morse for the constructive comments and suggestions. Below we respond point-by point to the comments:

**I do have one major concern that needs to be rectified. I am not sure but I think that it relates to the precipitation component of the ERA-Interim dataset. If you look at Figure 8b, you can see a very distinct artifact in the mapped data, especially in the inset.**

Response: We agree that the distinct artifact of precipitation 2000-2016 ERA-Interim dataset showed in the Figure 8b is a major concern, but we used it in the previous version to expand the time frame of the analysis. The results were consistent with the main conclusions that we made using University of Delaware 2000-2014, giving us confidence that these data could be used even with the artifact. However, we should have pointed out this artifact in the previous version of the manuscript to clarify this choice. For the new revision, we sought another source of precipitation and temperature data that would not have the artifact and would extend through the 2016 water year. To this end, we chose MERRA2, which has coarse resolution ( $0.625 \times 0.5$  grid), but it does not show a distinct artifact, and the main conclusions are similar to those we found using the ERA-Interim and University of Delaware datasets.

Minor comments:

**P1,L24: Delete "in magnitude"**

Response: these words have been deleted.

**P1, L34: Change "warming" to "increasing"**

Response: this change has been made.

**P2, L5: comma after (17°S to 31°S)**

Response: this change has been made.

**P2, L13: Change "related with" to "related to"**

Response: this change has been made.

**P2**, L16 and elsewhere: There is always a space between the value the "°" symbol for temperatures**

Response: we added a space after all "°" symbols.

P3, L4-6: Change "The Andes Mountains cross seven countries (Vene ... tina) along more than 8,000 km across a wide range of latitudes ... They represent ... and are the longest mountain chain..." to "The Andes Mountains cross seven countries (Vene ... tina) and a wide range of latitudes ... They represent ... and are the longest (8,000 km) mountain chain..."

Response: this change has been made.

**P3**, Figure 1: Please put a black box in the main map around the area that is "zoomed in to" in the inset, and thicker lines connecting the two.**

Response: this change is included in a new version of Figure 1.

**P4, L5: Comma after "area"**

Response: comma was added after area.

**P4, L7: Change "In these latitudes" to "At these latitudes"** *Response: this change has been made.*

P5, Figure 2 (and other similar composed figures): It is not obvious that the West/East relates to continental divide. Perhaps you should include this in the caption. How about changing "Mean monthly hydroclimate variables by latitude..." to "Mean monthly hydroclimate variables by location with respect to the continental divide and latitude..."

Response: we included this suggestion in all of the relevant captions.

**P5**, L16: Change "continuing in real time monthly" to "continues in real-time monthly increments"**

Response: this change has been made.

**P5**, L18: Given the different resolutions of these datasets and the 0.5° working resolution of the paper, please include a line about how the data were combined and the interpolation of the ERA-Interim data. Was there any fitting of one to the other? *Response: We did not interpolate ERA-Interim or MERRA2 datasets; rather we resampling the spatial resolution to 500m to fit with MODIS resolution. We included an explanation about this in the methods section.*

P6, L8: Change "surface temperature regions (SST)" to "surface temperature (SST)" regions"

Response: this change has been made.

**P6, L15: Please define NOAA here and delete from P14, L8.**

Response: NOAA was defined in this page.

**P6, L22: Change "America" to American"**

Response: this change has been made.

P6, L38: R is used for all analysis, so move much of the text from P7, L10-11, to here. Change "Saavedra et al. (2017). For the remaining pixels we used the" to "Saavedra et al. (2017). For the remaining pixels we conducted geo-statistical analysis on the data using the statistical computing R software (R core Team, 2013). We used the" *Response: this change has been made.*

**P7, L14 and elsewhere: the parameters r and r2 should be italicized. This affects text throughout the main document, tables and figures.**

Response: this change has been made throughout the manuscript.

**P7, L26: Delete "show"**

Response: this word was deleted.

**P8, L17: Comma after "29°S"**

Response: this change has been made.

**P10, L4: change "common range of change values are in the range of" to "common change values are on the order of"**

Response: this change has been made.

**P13**, Figure 6: The line symbol for the continental divide is not visible in the legend or the figure.**

Response: a new version of Figure 6 is included.

P15, L11: Change "is represented as a line for significant (not significant) in thick, (SP,S-), (thin, lines." To "is represented where significant as a thick line (SP,S-) and a thin line where not significant (SP,NS+)."

Response: this change has been made.

**P15, L12: Change "of UDELv4 and ERA-Interim datasets" to "from UDELv4 to ERA-Interim datasets"**

Response: this change has been made and includes the new MERRA2 dataset.

**P15, Figure 7: Lines for N3 and MEI are indistinguishable. Traces are also very pixelated. I hope that final line art diagrams for this manuscript will be vector based and smooth. Classes of Niño/Niña events are difficult to distinguish and need more contrast.**

Response: a new version of Figure 7 is included to address this comment.

**P16**, Figure 8: Obvious artifact in (b). Can't see the line symbol for the continental divide in the figure or legend**

Response: this change has been made.

**P19, Figure 10: Obvious artifact in (a) and (b).**

Response: a new version of Figure 8 was included with MERRA2 dataset.

**P20**, Figure 11: (b), is this PRI or RI? Please coordinate with caption. (c) part of figure title is missing. (d) is this RI or PRI?**

Response: a new version of Figure 11 is included to address this comment.

**P22, L16: Effect of apparent precipitation anomaly on results?**

Response: we changed the precipitation dataset to avoid the anomaly.

**P23, L28: What is the effect of the apparent precipitation-related anomaly on this conclusion?**

Response: we changed the precipitation dataset to avoid the anomaly.

---

## Author Response (AR4)

We thank the editor Peter Morse and respond point-by point to the comments:

Minor comments:

**P1, L11: Here you say the Andes are 7,000 km in length, but on P3, L6, you say 8,000 km. I think most estimates are closer to 7,000 km.**
*Response: we change from 8,000 to 7,000 km.*

**P1, L18: Delete "Interim". Should ENSO, SAM, and PDO be spelled out, as you did for MODIS? Similarly, Should MERRA2 also be spelled out? I think so given our broad readership.**
*Response: these words have been deleted. ENSO, SAM, and PDO were defined in this page.*

**Throughout: As before, there is always a space between the value the ""°"" symbol for temperatures, but not for position (latitude or longitude). I think you may have carried out a blanket search-and-replace that introduced some errors.**
*Response: we put a space between the value of temperature and the "°" but not for position.*

**Figure 6: The line symbol for the continental divide is still not clearly visible to me in the legend or the figure. Can you make the lines thicker?**
*Response: we included a thicker line.*

**P15, L11: Change "think" to "thick"**
*Response: this change has been made.*

**P22, L3: Delete "up front"**
*Response: these words were deleted.*

**P24, L17: Change "For climate data, we tested several different gridded products" to "During the development of this paper, we tested several different gridded climate data products"**
*Response: this change has been made.*